# NoiseSDF2NoiseSDF: Learning Clean Neural Fields from Noisy Supervision

## Abstract

Reconstructing accurate implicit surface representations from point clouds remains a challenging task, particularly when data is captured using low-quality scanning devices. These point clouds often contain substantial noise, leading to inaccurate surface reconstructions. Inspired by the Noise2Noise paradigm for 2D images, we introduce NoiseSDF2NoiseSDF, a novel method designed to extend this concept to 3D neural fields. Our approach enables learning clean neural SDFs from noisy point clouds through noisy supervision by minimizing the MSE loss between noisy SDF representations, allowing the network to implicitly denoise and refine surface estimations. We evaluate the effectiveness of NoiseSDF2NoiseSDF on benchmarks, including the ShapeNet, ABC, Famous, and Real datasets. Experimental results demonstrate that our framework significantly improves surface reconstruction quality from noisy inputs.

## 1 Introduction

Learning from imperfect targets (Zhang & Sabuncu, 2018; Han et al., 2018; Lehtinen et al., 2018; Bora et al., 2018; Zhu et al., 2017) is a fundamental challenge in machine learning, particularly when obtaining clean labels is impractical or unfeasible. In image processing, the pioneering work of Noise2Noise (N2N) (Lehtinen et al., 2018) demonstrated that image restoration could effectively be achieved by observing multiple corrupted instances of the same image. Specifically, N2N leverages the principle that pixel values at identical coordinates in different noisy images ideally represent the same underlying true signal. Then, the model learns to restore clean images by minimizing a simple loss, such as mean squared error (MSE), between noisy observations, as shown in Figure 1 (a).

Extending N2N principles to 3D point clouds (Hermosilla et al., 2019; Ma et al., 2023), however, poses inherent limitations due to their unstructured nature. Unlike images organized on regular grids, point clouds exhibit deviations across all spatial coordinates without the benefit of a stable reference framework. This fundamental difference renders a direct extension of N2N impractical. Standard loss functions such as MSE prove ineffective, resorting to specialized loss functions like Earth Mover's Distance (EMD) to capture only soft geometric correspondences in point cloud data, see Figure 1 (b).

Recent advances in 3D shape surface reconstruction have introduced neural fields, such as neural Signed Distance Function (SDF) (Park et al., 2019; Mescheder et al., 2019; Cui et al., 2024; Zhu et al., 2024), which are capable of predicting continuous SDF values for any given query 3D coordinate. Our key observation is that neural SDF, which encodes the 3D shape as an SDF mapping from 3D coordinates to scalar distance values, exhibits a conceptual parallel to the mapping between pixel coordinates and pixel intensities in 2D images, as shown in Figure 1 (c). Building on this analogy, we hypothesize that neural SDFs can be denoised by using noisy SDF observations with the same MSE loss strategy inspired by the N2N principle in image restoration.

In contrast to paired noisy 2D images that can be readily acquired from cameras, 3D neural fields are typically learned through neural networks. Because image coordinates are discrete and finite, simply following the image-level N2N approach and denoising the SDF values only at specified coordinates, cannot ensure that the denoised SDF field remains complete and continuous over an infinite coordinate domain. To address the above limitation, we propose training a neural field network with denoising capability to predict clean SDF values without being restricted to specific query

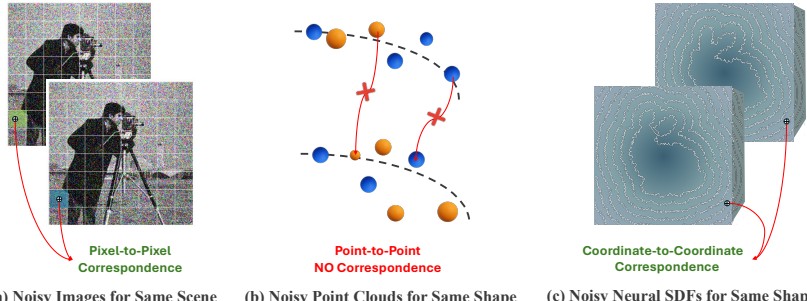

**Figure 1:** Comparison of coordinate correspondences: (a) Pixel coordinates represent correspondences between two noisy images of the same scene. In contrast, (b) point coordinates do not exhibit correspondences between two noisy point clouds of the same shape. (c) SDF coordinates establish correspondences between two noisy neural fields representing the same shape.

coordinates. Specifically, we introduce NoiseSDF2NoiseSDF, a novel approach that employs noisy-target supervision within neural SDFs to enable robust surface reconstruction from noisy 3D point clouds. The denoising network takes independently corrupted point clouds as input and predicts the underlying clean SDF values. Instead of using clean SDFs as ground truth, we employ another noisy neural SDF, generated by off-the-shelf point-to-SDF methods, as the supervision target. We then minimize the discrepancy between the predicted SDF output and the noisy SDF target using MSE loss. Through this process, the network learns to suppress noise and improve consistency across SDF values, resulting in clean neural representations.

To evaluate the effectiveness of NoiseSDF2NoiseSDF, we conduct comprehensive experiments across benchmark datasets, including ShapeNet (Chang et al., 2015), ABC (Koch et al., 2019), Famous (Erler et al., 2020), and Real (Erler et al., 2020). Our experimental results demonstrate that neural SDFs can indeed be denoised effectively by employing MSE loss directly between their noisy representations. This finding confirms our central hypothesis: neural SDFs can learn to produce cleaner outputs simply by observing and minimizing discrepancies among noisy neural fields, effectively extending the N2N paradigm into the domain of 3D shape surface reconstruction.

## 2 RELATED WORK

**Noise2Noise (N2N)** (Lehtinen et al., 2018) has significantly influenced recent image denoising. By leveraging pairs of noisy observations of the same scene, N2N learns to predict one noisy realization from another via pixel-wise correspondence. Subsequent methods like Noise2Void (Krull et al., 2019), Noise2Self (Batson & Royer, 2019) employ blind-spot masking techniques, training models directly on individual noisy images without pairs. Noise2Same (Xie et al., 2020) derives self-supervised loss bounds to eliminate the blind-spot restriction altogether. Self2Self (Quan et al., 2020) and Neighbor2Neighbor (Huang et al., 2021) exploit internal image redundancy, employing dropout or pixel resampling to train on single noisy observations without explicit noise modeling. Noisier2Noise (Moran et al., 2020) extends N2N to explicitly introduce additional synthetic noise, learning to map noisier images back to their original noisy versions.

Recent work has extended the N2N paradigm to the 3D domain, particularly to point clouds (Hermosilla et al., 2019; Wei, 2025; Ma et al., 2023; Wang et al., 2024). TotalDenoising (Hermosilla et al., 2019) and N2NM (Ma et al., 2023) employ soft local geometric correspondences with Earth Mover's Distance (EMD) loss to align noisy point clouds with the underlying surface. However, these methods are unable to establish exact point matches.

To the best of our knowledge, no prior work has applied the N2N paradigm to the domain of 3D neural fields. We are the first to exploit the structural similarities between neural fields and images by proposing an N2N denoising framework for 3D SDFs using a simple MSE loss, which enables direct SDF matches.

**Implicit Surface Reconstruction.** Overfitting-based methods optimize a neural implicit function for a single shape through intensive test-time optimization. They often achieve high geometric fi-

delity on that specific object but lack generalization to new shapes. For example, SAL (Atzmon & Lipman, 2020), SALD (Atzmon & Lipman, 2021), and Sign-SAL (Zhao et al., 2021) use point proximity and self-similarity cues. Gradient regularization techniques like IGR (Gropp et al., 2020), DiGS (Ben-Shabat et al., 2022), and Neural-Pull (Ma et al., 2020) improve stability and detail. Extensions such as SAP (Peng et al., 2021), LPI (Chen et al., 2022), and Implicit Filtering-Net (Li et al., 2024) enhance reconstruction under sparse sampling and complex geometry. Neural-Singular-Hessian (Wang et al., 2023b) pushes single-shape overfitting by leveraging a Hessian-based regularizer to achieve surface recovery. While accurate, these methods are typically sensitive to noise. Robust variants (e.g., SAP (Peng et al., 2021), PGR (Lin et al., 2022), Neural-IMLS (Wang et al., 2023a), N2NM (Ma et al., 2023) and LocalN2NM (Chen et al., 2024)) address this via smoothing, denoising priors, or self-supervision.

Data-driven methods learn from collections of shapes, allowing the model to infer implicit surfaces for previously unseen instances with efficient inference. For instance, global-latent methods, such as OCCNet (Mescheder et al., 2019), IM-NET (Chen, 2019), and DeepSDF (Park et al., 2019), encode entire shapes into fixed-length global latent codes. Local-based methods improve expressiveness by operating at finer scales. Grid-based approaches divide space into cells and learn small implicit functions per cell (ConvOccNet (Peng et al., 2020b), SSRNet (Peng et al., 2020a), Local Implicit Grid (Genova et al., 2020), Deep Local Shapes (Chabra et al., 2020)). Patch-based methods segment point clouds into local regions and learn shared atomic representations (PatchNets (Tretschk et al., 2020), POCO (Boulch & Marlet, 2022), neighborhood-based (Jiang et al., 2021)). Hybrid methods combine global context with local detail. For instance, IF-Nets (Chibane et al., 2020) and SG-NN (Dai et al., 2020) integrate local features within hierarchical representations. P2S (Qi et al., 2017) and PPSurf (Erler et al., 2024) use dual-branch networks to predict SDFs. Recent transformer-based models (ShapeFormer (Yan et al., 2022), 3DILG (Zhang et al., 2022), 3DS2V (Zhang et al., 2023), LaGeM (Zhang & Wonka, 2025)) leverage self-attention for long-range structure modeling. Since these methods are trained using ground-truth SDFs, their performance degrades when input point clouds are sparse or noisy. In this work, we demonstrate that clean neural fields can be learned under noisy supervision, enabling robust surface reconstruction from corrupted inputs.

## 3 PRELIMINARIES

In *Noise2Noise* (Lehtinen et al., 2018), the key idea is that given multiple noisy observations of the same underlying clean image, the pixel intensities at the same spatial coordinates are expected to share the same statistical properties. Formally, consider an image domain $\mathbf{X} \subset \mathbb{R}^2$, and let $y_1, y_2, \ldots, y_n$ be noisy observations of the same underlying clean image taken at different instances. For any pixel coordinate $x \in \mathbf{X}$, the pixel intensities $y_1(x), y_2(x), \ldots, y_n(x)$ are samples drawn from a distribution centered around the true pixel value at that location, perturbed by independent, zero-mean noise. The core insight of Noise2Noise is that even in the presence of such noise, the expectation of the noisy pixel values converges to the true signal:

$$\mathbb{E}[y_i(x)] = y(x), \quad \forall i \in \{1, 2, \ldots, n\}, \tag{1}$$

where $y_i(x)$ is the observed pixel value at coordinate $x$ in the $i$-th noisy image, and $y(x)$ is the true underlying pixel value at that coordinate. This property enables training a neural network purely on noisy data, using other noisy images as supervision.

Let $f_\theta$ denote a denoising network parameterized by $\theta$, which was initialized using ImageNet pre-training (He et al., 2015), and let $x \in \mathbf{X}$ represent a spatial query coordinate. The network is designed to predict pixel intensities given a noisy image and the query coordinate. The prediction is written as:

$$\hat{y}(x \mid y_i) = f_\theta(y_i, x), \tag{2}$$

where $y_i$ is the noisy input image, $x$ is the queried pixel location, and $\hat{y}(x \mid y_i)$ is the predicted pixel intensity at $x$. The model is trained to minimize the expected squared error between the predicted pixel value and the corresponding pixel value in another independent noisy observation. The loss function is:

$$\mathcal{L}(\theta) = \mathbb{E}_{y_1, y_2 \sim p(y|y), x \sim \mathcal{U}(\mathbb{R}^2)} \left[ \|\hat{y}(x \mid y_1) - y_2(x)\|^2 \right], \tag{3}$$

where $y_1, y_2$ are independent noisy observations of the same clean image, and $x \in \mathbf{X}$ is sampled uniformly from the image domain. After training, given a noisy image, $f_\theta$ can predict a denoised version.

# 4 METHOD

Our proposed method investigates whether clean neural fields can be effectively learned by observing their noisy counterparts. Drawing inspiration from Noise2Noise, where noisy images directly serve as inputs and targets, we adapt this principle to learning neural fields from noisy point cloud data. In contrast to the direct usage of noisy images as input in traditional Noise2Noise setups, we employ a neural network conditioned on a noisy point cloud to predict neural SDFs at given query coordinates. Rather than utilizing clean SDFs as supervision, our approach leverages noisy neural fields at identical coordinates derived from another independently noisy version of the same underlying shape. This ensures one-to-one correspondence between the predicted and target neural fields, allowing effective noise suppression through direct MSE loss minimization.

## 4.1 NOISESDF2NOISESDF

Formulating *Noise2Noise* (Lehtinen et al., 2018) to Signed Distance Functions (SDFs) introduces new opportunities for denoising in 3D spaces. Unlike unstructured point clouds, SDFs represent 3D geometry in a structured and continuous manner, mapping each spatial coordinate $q \in \mathbb{R}^3$ to its signed distance from the surface of an underlying object. This continuity ensures that, for the same query coordinate across multiple noisy observations derived from the same shape, the SDF values should remain statistically consistent. Let $p_1, p_2, \ldots, p_n$ be noisy point cloud observations of the same underlying 3D shape, and let $s_1, s_2, \ldots, s_n$ be their corresponding noisy SDFs. Given a noisy point cloud $p_i$ and a query coordinate $q$, a neural network $f_\theta$, parameterized by $\theta$, is trained to predict the SDF value $\hat{s}(q \mid p_i)$ at the queried location:

$$\hat{s}(q \mid p_i) = f_\theta(p_i, q). \tag{4}$$

The structured nature of SDFs enables the network to learn smooth and continuous surface representations, even from sparse or noisy inputs. This makes SDFs advantageous over unordered point clouds for tasks like 3D denoising and reconstruction.

**Training Objective.** The model is trained by minimizing the expected squared error between the predicted SDF value from one noisy observation and the SDF value at the same query location in another noisy observation of the same shape. The loss function is defined as:

$$\mathcal{L}(\theta) = \mathbb{E}_{p_1, p_2 \sim p(p|s), q \sim \mathcal{U}(\mathbb{R}^3)} \left[ \|\hat{s}(q \mid p_1) - s_2(q \mid p_2)\|^2 \right], \tag{5}$$

where $p_1, p_2$ are independent noisy point cloud observations sampled from the same underlying shape $s$, and $s_2(q \mid p_2)$ is the noisy SDF value at coordinate $q$ associated with noisy point cloud $p_2$.

This formulation takes advantage of the continuous nature of SDFs, which, unlike point clouds, allows for consistent supervision across noisy samples even if the raw point distributions are unstructured. By learning to map noisy coordinates to structured SDF representations, the neural network effectively filters noise, yielding a refined and more accurate 3D representation of the surface.

## 4.2 IMPLEMENTATION

Our framework is illustrated in Figure 2. The process begins with sampling sparse, noisy point clouds from a watertight surface. During training, a pair of noisy point clouds is randomly selected: one is processed through the neural SDF network to predict approximate clean SDF values for the underlying 3D shape. Simultaneously, a point-to-SDF method is applied to generate a noisy SDF target, which serves as noisy supervision during the denoising phase. For each query point, the corresponding SDF values from these two representations are extracted and compared using the Mean Squared Error (MSE) loss function. This loss is then utilized to update the weights of the neural SDF network during denoising.

**Point Sampling.** We first normalize the watertight meshes into a unit cube, then sample points from the surfaces to obtain the original point cloud $p$. Following the Noise2Noise protocols (Lehtinen et al., 2018; Ma et al., 2023), we apply zero-mean Gaussian noise to generate noisy point cloud pairs. The query point set consists of $50\%$ near-surface points and $50\%$ uniformly sampled points from the unit cube. To reduce dependency on the original clean surface, we directly use the two input noisy point clouds as the near-surface query points. Additionally, we uniformly sample $N$ points within the cube as spatial query points.

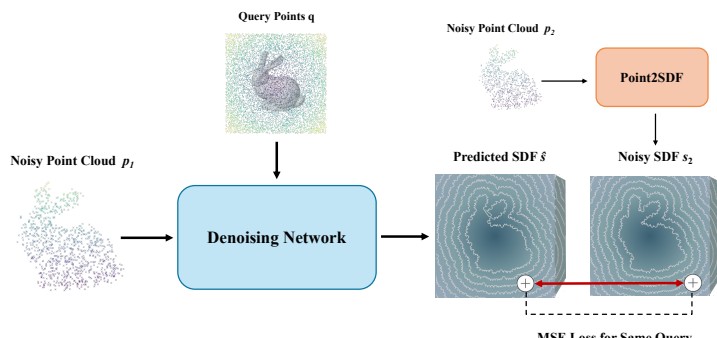

**Figure 2:** The training pipeline of the NoiseSDF2NoiseSDF framework. Given two independent noisy point clouds $p_1$ and $p_2$ of the same underlying shape, $p_1$ is fed into the denoising network to predict a smoothed SDF $\hat{s}$, while $p_2$ is passed through a Point2SDF network to generate a noisy SDF $s_2$. Both SDFs are evaluated at a shared set of query points $q$, and their mean squared error is used to update the denoising network weights.

**Denoising Network.** Our SDF prediction network is built on 3DS2V (Zhang et al., 2023). Initially, a noisy point cloud $p_1$ is sampled and transformed into positional embeddings, which are then encoded into a set of latent codes through a cross-attention module. Subsequently, self-attention is applied to aggregate and exchange information across the latent set, enhancing feature integration. A cross-attention module then computes interpolation weights for the query point $q$. These interpolated feature vectors are processed through a fully connected layer to predict SDF values. The network weights are initialized following Zhang et al. (2023) to accelerate convergence.

**Noisy Target.** Given another paired noisy point cloud $p_2$, a Point2SDF method is required to predict noisy SDF values $s_2$ from it. In this implementation, we use the 3DS2V (Zhang et al., 2023) with frozen parameters. Since it is not trained in a denoising setting, noisy inputs produce noisy SDF outputs. This network accepts as input a noisy point cloud $p_2$ and a query point $q$, producing the corresponding noisy SDF scalar value at $q$. To ensure that all SDF targets are consistently noisy, we freeze its parameters during this process.

**Inference**. After training, only the Denoising Network is used for inference. Given a noisy point cloud, the network predicts the denoised SDF values in a *single* forward pass. The clean underlying surface is then recovered from the denoised SDF using Marching Cubes (Lorensen & Cline, 1987).

## 5 EXPERIMENT

### 5.1 TRAINING DETAILS

For optimization, we used the AdamW optimizer (Loshchilov & Hutter, 2019) with a fixed learning rate of $1 \times 10^{-4}$. For resource usage, we trained on three Nvidia A100 GPUs with a batch size of 32 per GPU, taking approximately 15 hours for the ShapeNet dataset and 2.5 hours for the ABC dataset. We sampled 2048 points from watertight meshes as the initial point cloud. Following the N2NM (Ma et al., 2023), we applied Gaussian noise with standard deviations of 1%, 2%, online to generate noisy and sparse point cloud pairs. Additionally, we sampled 8192 query points online. The noise magnitude is defined with respect to both the point-cloud bounding-box size and the point density. For a fixed numeric noise level, a smaller bounding box amplifies the relative impact of the perturbation. All point clouds are normalized to the cubes $[-0.5, 0.5]^3$ or $[-1, 1]^3$. Furthermore, sparser point sets are more susceptible to noise. With only 2048 points, noise levels of 0.01 and 0.02 constitute *severe* perturbations irrespective of the bounding-box scale.

### 5.2 DATASETS AND METRICS

We evaluate our NoiseSDF2NoiseSDF on ShapeNet following Zhang et al. (2023). To assess denoising effectiveness and surface reconstruction quality, we use metrics, including Intersection-over-Union (IoU), Chamfer Distance, F1 Score, and Normal Consistency (NC). IoU was computed based on occupancy predictions over densely sampled volumetric points. Following methods (Li et al.,

**Table 1:** Comparison of 3DS2V (Zhang et al., 2023) and Ours on ShapeNet with Gaussian noise $\sigma = 0.01$.

| Category | IoU ↑ | | | NC ↑ | | | Chamfer ↓ | | | F-Score ↑ | | |
|---|---|---|---|---|---|---|---|---|---|---|---|---|
| | 3DS2V | Ours | Δ | 3DS2V | Ours | Δ | 3DS2V | Ours | Δ | 3DS2V | Ours | Δ |
| table | 0.879 | **0.922** | +0.043 | 0.930 | **0.976** | +0.046 | 0.013 | **0.012** | +0.001 | 0.991 | **0.992** | +0.001 |
| car | 0.946 | **0.959** | +0.013 | 0.890 | **0.908** | +0.018 | 0.022 | **0.020** | +0.002 | 0.925 | **0.925** | +0.000 |
| chair | 0.887 | **0.921** | +0.034 | 0.937 | **0.966** | +0.029 | 0.014 | **0.013** | +0.001 | 0.986 | **0.986** | +0.000 |
| airplane | 0.884 | **0.931** | +0.047 | 0.939 | **0.972** | +0.033 | 0.010 | **0.008** | +0.002 | 0.997 | **0.997** | +0.000 |
| sofa | 0.946 | **0.964** | +0.018 | 0.943 | **0.974** | +0.031 | 0.014 | **0.012** | +0.002 | 0.986 | **0.987** | +0.001 |
| rifle | 0.821 | **0.910** | +0.089 | 0.869 | **0.960** | +0.091 | 0.009 | **0.007** | +0.002 | 0.997 | **0.998** | +0.001 |
| lamp | 0.826 | **0.894** | +0.068 | 0.904 | **0.952** | +0.048 | 0.011 | **0.009** | +0.002 | 0.989 | **0.989** | +0.000 |
| mean | 0.884 | **0.929** | +0.045 | 0.916 | **0.958** | +0.042 | 0.0132 | **0.0113** | +0.0019 | 0.981 | **0.986** | +0.005 |

2024; Ma et al., 2020), we sampled $1 \times 10^5$ points from the reconstructed and ground-truth surfaces to compute the Chamfer Distance and F1 Score.

To further evaluate the generalization ability of our approach, we trained the model on the ABC training set (Koch et al., 2019) and tested it on the ABC test set, as well as the Famous (Erler et al., 2020) and Real (Erler et al., 2020) datasets. Importantly, neither of these datasets was used during 3DS2V's training, so they can be regarded as out-of-distribution. Our model is trained solely with noisy targets generated by 3DS2V, without relying on any clean ground-truth from these datasets. We utilized the preprocessed datasets and data splits provided by Erler et al. (2020; 2024). We reported evaluation metrics including Normal Consistency, Mesh Normal Consistency, Chamfer Distance, and F1 Score. All metrics reported above are evaluated on the reconstructed meshes. We excluded IoU from this benchmark because, under severe noise, many reconstructed meshes become non-watertight or heavily degenerated, making it infeasible to assign reliable inside/outside labels and rendering the IoU metric unreliable.

## 5.3 RESULTS ON SHAPENET

To verify our hypothesis that clean neural fields can be learned from noisy supervision, we comprehensively compare our method with 3DS2V (Zhang et al., 2023) on the seven largest ShapeNet (Chang et al., 2015) subsets, following its experimental setup. Since our method is trained using 3DS2V as the denoising network with only noisy supervision, achieving superior performance over 3DS2V would therefore provide strong evidence supporting our hypothesis. We adopt the officially released 3DS2V model as our baseline. 3DS2V was trained with pairs of clean point clouds and clean SDFs for supervision, and it was not exposed to noisy inputs paired with clean SDF. To ensure a fair evaluation, we adopt the same data splits and preprocessing procedures as 3DS2V. Our NoiseSDF2NoiseSDF is trained under noisy supervision, meaning that we do not use any paired noisy point clouds and clean SDFs throughout the entire training process. We report evaluation results for each subset at noise levels of 0.01 (Table 1) and 0.02 (Table 2) and show visualization results in Figure 3. Under lower corruption ($\sigma = 0.01$), our method outperforms 3DS2V across all evaluation metrics. For example, the mean results show an IoU increase of 0.045 (5.1%), a Normal Consistency improvement of 0.042 (4.6%), and a reduction in Chamfer Distance from 0.0132 to 0.0113. At the higher corruption level ($\sigma = 0.02$), our approach remains the robust and stable with the better mean metrics that surpass the baseline. For instance, the mean results show an IoU increase of 0.08 (16.3%), a Normal Consistency improvement of 0.135 (18.4%), and an F-Score increase of 0.047 (6%). The results demonstrate that our method outperforms the baseline, 3DS2V, under both corruption levels and confirm our central idea that it is possible to learn to clean neural fields from noisy supervision.

## 5.4 RESULTS ON ABC, FAMOUS, AND REAL

We compared results on the ABC, Famous, and Real test datasets provided by P2S (Erler et al., 2020). Specifically, we evaluated data-driven methods P2S, PPSurf (Erler et al., 2024), POCO (Boulch & Marlet, 2022), and NKSR (Huang et al., 2023), known for their strong noise resilience in point cloud data. Note that, except for our method, *all other approaches are trained on the ABC dataset directly using ground-truth noisy–clean pairs*. For these methods, we used their officially released pretrained models. Quantitative results are reported in Table 3, and qualitative mesh

**Table 2:** Comparison of 3DS2V (Zhang et al., 2023) and Ours on ShapeNet with Gaussian noise $\sigma = 0.02$.

| Category | IoU ↑ | | | NC ↑ | | | Chamfer ↓ | | | F-Score ↑ | | |
|---|---|---|---|---|---|---|---|---|---|---|---|---|
| | 3DS2V | Ours | Δ | 3DS2V | Ours | Δ | 3DS2V | Ours | Δ | 3DS2V | Ours | Δ |
| table | 0.528 | **0.591** | +0.063 | 0.765 | **0.912** | +0.147 | 0.029 | **0.028** | +0.001 | 0.792 | **0.859** | +0.067 |
| car | 0.434 | **0.491** | +0.057 | 0.715 | **0.787** | +0.072 | **0.040** | 0.044 | -0.004 | 0.669 | **0.688** | +0.019 |
| chair | 0.463 | **0.530** | +0.067 | 0.729 | **0.868** | +0.139 | 0.034 | 0.035 | -0.001 | 0.694 | **0.721** | +0.027 |
| airplane | 0.465 | **0.536** | +0.071 | 0.719 | **0.856** | +0.137 | 0.025 | **0.022** | +0.003 | 0.830 | **0.899** | +0.069 |
| sofa | 0.355 | **0.425** | +0.070 | 0.769 | **0.866** | +0.097 | **0.036** | 0.038 | -0.002 | 0.667 | **0.677** | +0.010 |
| rifle | 0.625 | **0.781** | +0.156 | 0.691 | **0.891** | +0.200 | 0.021 | **0.014** | +0.007 | 0.887 | **0.968** | +0.081 |
| lamp | 0.572 | **0.649** | +0.077 | 0.744 | **0.896** | +0.152 | 0.026 | **0.025** | +0.001 | 0.825 | **0.880** | +0.055 |
| mean | 0.492 | **0.572** | +0.080 | 0.733 | **0.868** | +0.135 | 0.030 | **0.029** | +0.001 | 0.766 | **0.813** | +0.047 |

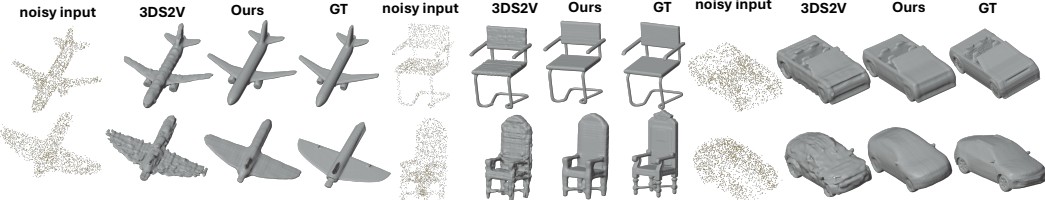

**Figure 3:** Comparison on the ShapeNet dataset. The first row shows Gaussian noise with $\sigma = 0.01$, and the second row with $\sigma = 0.02$. Ours produces smoother reconstructions, better aligning with the underlying surfaces compared to baseline.

reconstructions are shown in Figure 4. Across noise levels, our method achieves strong performance on mean NC and Mesh NC, indicating coherent geometry and smooth surfaces; this is also evident in the visual reconstructions (Figure 4). At the 0.01 noise level, the ABC and Famous datasets achieve NC scores of 0.865 and 0.831, respectively. When the noise level increases to 0.02, our method yields the best mean Mesh NC and strong mean NC among them. These results demonstrate that even when trained with noisy supervision and without ground-truth, our method can achieve performance competitive with state-of-the-art data-driven approaches.

### 5.5 QUANTITATIVE RESULTS SUPPLEMENTARY

We compare our method with representative overfitting-based approaches, such as SAP-O (Peng et al., 2021) and PGR (Lin et al., 2022). These methods train separate networks for each test shape, require long inference times, and lack generalization to unseen shapes. We adopted the training configurations recommended or set as default in their respective works. Table 4 shows that SAP-O and PGR can sometimes achieve lower Chamfer Distance and higher F1, but our approach consistently outperforms them in NC and Mesh NC. At the 0.01 noise level, our method achieves average NC/Mesh NC/F-Score values of 0.847/0.027/0.945, all of which are the best scores. At the 0.02 noise level, our method also demonstrates strong performance, with higher NC and Mesh NC, as well as competitive Chamfer Distance and F-Score.

We further include a comparison with N2NM (Ma et al., 2023). For each shape, N2NM was trained with 200 noisy samples, which resulted in long inference times (46 minutes) per shape. To ensure a fair comparison, we conducted experiments on the Famous dataset following the N2NM setup (Zhou et al., 2024) and incorporated test-time optimization (TTO) into our method with the same number of noisy samples. As shown in Table 5, our method combined with TTO demonstrated significant self-improvement, outperformed N2NM under medium noise conditions, and achieved comparable results under maximum noise, while delivering inference that is orders of magnitude faster.

### 5.6 ABLATION STUDY

Our ablation experiments use the "Chair" subset of ShapeNet, with 6271 models for training, 169 for validation, and 338 for testing.

**Table 3:** Comparison of P2S (Erler et al., 2020), PPSurf (Erler et al., 2024), POCO (Boulch & Marlet, 2022), NKSR (Huang et al., 2023), and Ours on six noisy test datasets. The best two performances are highlighted.

| Dataset | NC ↑ | | | | | Mesh NC ↓ | | | | | Chamfer ↓ | | | | | F-Score ↑ | | | | |
|---|---|---|---|---|---|---|---|---|---|---|---|---|---|---|---|---|---|---|---|---|
| | P2S | PPSurf | POCO | NKSR | Ours | P2S | PPSurf | POCO | NKSR | Ours | P2S | PPSurf | POCO | NKSR | Ours | P2S | PPSurf | POCO | NKSR | Ours |
| ABC ($\sigma = 0.01$) | 0.790 | 0.770 | 0.864 | 0.800 | **0.865** | 0.330 | 0.059 | 0.025 | 0.039 | **0.024** | 0.017 | 0.017 | **0.014** | 0.018 | 0.015 | 0.919 | 0.935 | **0.941** | 0.927 | 0.938 |
| ABC ($\sigma = 0.02$) | 0.753 | 0.728 | **0.848** | 0.727 | 0.812 | 0.381 | 0.061 | 0.020 | 0.058 | **0.018** | 0.027 | **0.022** | 0.019 | 0.026 | 0.032 | 0.852 | 0.870 | **0.898** | 0.801 | 0.724 |
| Famous ($\sigma = 0.01$) | 0.771 | 0.761 | 0.825 | 0.775 | **0.831** | 0.268 | 0.053 | 0.026 | 0.032 | **0.025** | 0.017 | **0.015** | 0.017 | 0.017 | 0.016 | 0.928 | 0.959 | **0.962** | 0.943 | 0.941 |
| Famous ($\sigma = 0.02$) | 0.727 | 0.728 | **0.785** | 0.703 | 0.767 | 0.328 | 0.054 | **0.023** | 0.054 | 0.024 | 0.022 | **0.020** | 0.022 | 0.026 | 0.032 | 0.868 | **0.899** | 0.871 | 0.810 | 0.726 |
| Real ($\sigma = 0.01$) | 0.789 | 0.776 | 0.845 | 0.779 | 0.845 | 0.177 | 0.057 | 0.032 | 0.038 | **0.031** | 0.016 | 0.016 | **0.014** | 0.015 | 0.015 | 0.946 | 0.954 | **0.964** | 0.955 | 0.956 |
| Real ($\sigma = 0.02$) | 0.734 | 0.745 | **0.803** | 0.700 | 0.793 | 0.269 | 0.053 | 0.024 | 0.059 | **0.020** | 0.021 | 0.022 | 0.025 | 0.029 | 0.026 | 0.877 | 0.876 | **0.930** | 0.822 | 0.809 |
| mean ($\sigma = 0.01$) | 0.783 | 0.769 | 0.844 | 0.785 | **0.847** | 0.258 | 0.056 | 0.028 | 0.036 | **0.027** | 0.017 | **0.014** | 0.016 | 0.017 | 0.015 | 0.931 | 0.949 | **0.962** | 0.930 | 0.945 |
| mean (all) | 0.761 | 0.751 | **0.828** | 0.747 | 0.819 | 0.292 | 0.056 | 0.025 | 0.047 | **0.024** | 0.020 | **0.019** | 0.019 | 0.022 | 0.023 | 0.898 | **0.916** | 0.915 | 0.876 | 0.849 |

**Table 4:** Comparison of SAP-O (Peng et al., 2021), PGR (Lin et al., 2022), and Ours on six noisy test datasets. The adaptive resolution in PGR can occasionally drop relatively low, resulting in artificially smooth meshes.

| Dataset | NC ↑ | | | Mesh NC ↓ | | | Chamfer ↓ | | | F-Score ↑ | | |
|---|---|---|---|---|---|---|---|---|---|---|---|---|
| | SAP-O | PGR | Ours | SAP-O | PGR | Ours | SAP-O | PGR | Ours | SAP-O | PGR | Ours |
| ABC ($\sigma = 0.01$) | 0.710 | 0.835 | **0.865** | 0.079 | 0.037 | **0.024** | 0.021 | 0.020 | **0.014** | 0.906 | 0.896 | **0.938** |
| ABC ($\sigma = 0.02$) | 0.622 | 0.778 | **0.812** | 0.095 | 0.065 | **0.018** | 0.026 | **0.026** | 0.032 | **0.824** | 0.815 | 0.724 |
| Famous ($\sigma = 0.01$) | 0.745 | 0.813 | **0.831** | 0.053 | 0.035 | **0.025** | 0.022 | 0.017 | **0.016** | 0.876 | 0.931 | **0.941** |
| Famous ($\sigma = 0.02$) | 0.614 | 0.755 | **0.767** | 0.104 | 0.064 | **0.024** | **0.023** | 0.024 | 0.032 | 0.849 | 0.834 | **0.726** |
| Real ($\sigma = 0.01$) | 0.683 | 0.827 | **0.845** | 0.097 | 0.032 | **0.031** | 0.023 | 0.015 | **0.015** | 0.902 | 0.956 | **0.956** |
| Real ($\sigma = 0.02$) | 0.595 | 0.756 | **0.793** | 0.122 | 0.062 | **0.020** | **0.025** | 0.026 | 0.026 | **0.841** | 0.824 | 0.809 |
| mean ($\sigma = 0.01$) | 0.713 | 0.825 | **0.847** | 0.076 | 0.035 | **0.027** | 0.022 | 0.017 | **0.015** | 0.895 | 0.928 | **0.945** |
| mean (all) | 0.661 | 0.794 | **0.819** | 0.092 | 0.049 | **0.024** | 0.023 | **0.021** | 0.023 | 0.866 | **0.876** | 0.849 |

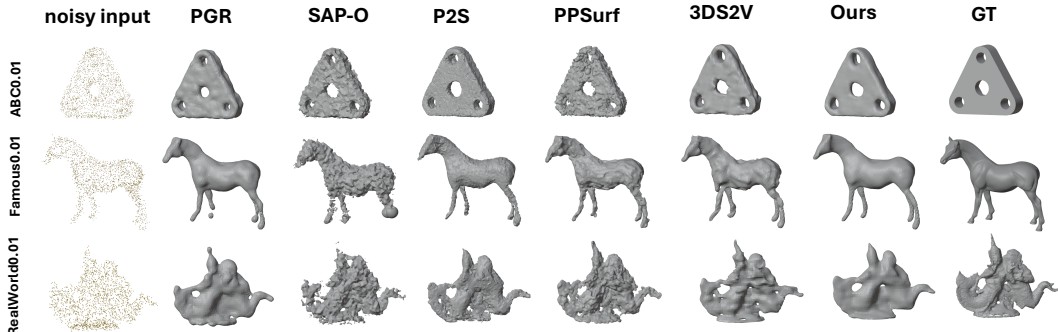

**Figure 4:** Visualization of surface reconstruction under Gaussian noise ($\sigma = 0.01$ and $0.02$), comparing two overfitting-based methods (PGR (Lin et al., 2022) and SAP-O (Peng et al., 2021)), three data-driven approaches (P2S (Erler et al., 2020), PPSurf (Erler et al., 2024), 3DS2V (Zhang et al., 2023)), and our proposed method. Results are shown for three benchmark datasets: ABC (Koch et al., 2019), Famous (Erler et al., 2020), and Real (Erler et al., 2020).

**Noisy vs. Clean Supervision.** In the context of Noise2Noise, clean supervision corresponds to Noise2Clean, where ground-truth SDFs are provided during training. For a fair comparison, we conducted experiments under identical training conditions (batch size, shapes numbers/batch, etc.) between our noisy and clean supervision with a noise level of 0.01. The results are reported in Table 6 (Clean as target). We observe that our noisy supervision achieves performance nearly equivalent to that of clean supervision, which aligns with the findings of Lehtinen et al. (2018). This supports our main hypothesis that cleaner outputs can be produced by observing noisy neural fields.

**Noisy Targets and Denoising Networks.** To validate that our framework can relax constraints on the architecture of the SDF prediction (i.e., Point2SDF), we conducted a study by replacing 3DS2V with 3DILG as the noisy targets. We further used 3DILG as the denoising network and performed same experiments. 3DILG (Zhang et al., 2022) encodes 3D shapes using irregular latent grids, whereas 3DS2V represents neural fields with a set of vectors. In Table 6, the results indicate that our method remains effective in learning clean SDFs from noisy supervision. These findings support the generalizability of our approach across different forms of noisy supervision and denoising networks.

**Table 5:** Comparison of N2NM and our method under medium noise (MedN) and maximum noise (MaxN) using Chamfer Distance on the Famous dataset.

| Method | MedN Chamfer ↓ | MaxN Chamfer ↓ | Inference time |
|---|---|---|---|
| N2NM | 0.0132 | **0.0231** | 2760 seconds |
| Ours | 0.0160 | 0.0320 | **0.05 seconds** |
| Ours-TTO | **0.0108** | 0.0252 | 20 seconds |

**Table 6:** Quantitative comparisons for different targets and denoising networks.

**(a)** Using 3DS2V as denoising network.

| Method | IoU ↑ | Chamfer ↓ | F-Score ↑ | NC ↑ |
|---|---|---|---|---|
| Baseline | 0.887 | 0.014 | 0.986 | 0.937 |
| Clean | 0.939 | 0.013 | 0.988 | 0.970 |
| Noisy(3DILG) | 0.903 | 0.014 | 0.980 | 0.950 |
| Noisy(3DS2V) | 0.927 | 0.013 | 0.986 | 0.966 |

**(b)** Using 3DILG as denoising network.

| Method | IoU ↑ | Chamfer ↓ | F-Score ↑ | NC ↑ |
|---|---|---|---|---|
| Baseline | 0.881 | 0.015 | 0.977 | 0.930 |
| Clean | 0.921 | 0.014 | 0.981 | 0.960 |
| Noisy(3DILG) | 0.907 | 0.014 | 0.979 | 0.953 |
| Noisy(3DS2V) | 0.913 | 0.014 | 0.978 | 0.962 |

**Denoising Network Training.** Fine-tuning only the fully connected layer offers virtually no benefit. Adding the cross-attention block introduces a clear gain. Fine-tuning the entire decoder achieves the better performance. We further evaluate the random initialization of the decoder and the entire network. Table 7 shows that initializing the decoder from 3DS2V does not improve performance but does accelerate convergence. Considering the trade-off between performance and training cost, we adopt the strategy of freezing the encoder while fine-tuning the decoder.

**Table 7:** Comparison of training strategies on the denoising network.

| Metric | Baseline | FC | FC+CA | Decoder | Decoder-RI | Network-RI |
|---|---|---|---|---|---|---|
| IoU | 0.887 | 0.884 | 0.922 | 0.927 | 0.927 | 0.929 |
| NC | 0.937 | 0.933 | 0.954 | 0.966 | 0.967 | 0.968 |
| Epochs | 0 | 10 | 15 | 30 | 350 | 800 |

**Noise Type.** Beyond standard zero-mean Gaussian noise, we evaluated three additional noise types, Uniform, Discrete, and Laplace noise, each applied at a fixed magnitude of $\sigma = 0.01$. Furthermore, to assess the impact of non-zero bias in Gaussian perturbations, we conducted experiments over the domain $[-1, 1]^3$ using means of $\mu = 0.005, 0.01$, and $0.02$. Comprehensive quantitative results are presented in Table 8. At $\sigma = 0.01$ on $[-1, 1]^3$, our model consistently shows denoising performance under Uniform and Discrete noise, with notable gains in both IoU and NC.

**Table 8:** Comparison between the 3DS2V (Zhang et al., 2023) and Ours under various noise types.

| Dataset | IoU ↑ | | | NC ↑ | | | Chamfer ↓ | | | F-Score ↑ | | |
|---|---|---|---|---|---|---|---|---|---|---|---|---|
| | 3DS2V | Ours | Δ | 3DS2V | Ours | Δ | 3DS2V | Ours | Δ | 3DS2V | Ours | Δ |
| Uniform ($\sigma = 0.01$) | 0.873 | **0.911** | +0.038 | 0.920 | **0.962** | +0.042 | 0.015 | **0.013** | +0.002 | 0.985 | **0.986** | +0.001 |
| Discrete ($\sigma = 0.01$) | 0.867 | **0.895** | +0.028 | 0.915 | **0.960** | +0.045 | 0.015 | **0.014** | +0.001 | 0.984 | **0.985** | +0.001 |
| Laplace ($\sigma = 0.01$) | **0.909** | 0.908 | -0.001 | 0.956 | **0.956** | +0.000 | 0.014 | **0.014** | +0.000 | 0.985 | **0.985** | +0.000 |
| Gaussian ($\sigma = 0.01$) | 0.887 | **0.927** | +0.040 | 0.937 | **0.966** | +0.029 | 0.014 | **0.013** | +0.001 | 0.986 | **0.986** | +0.000 |
| Gaussian ($\sigma = 0.01, \mu = 0.005$) | 0.860 | **0.881** | +0.021 | 0.942 | **0.964** | +0.022 | 0.017 | **0.016** | +0.001 | 0.982 | **0.984** | +0.002 |
| Gaussian ($\sigma = 0.01, \mu = 0.01$) | 0.798 | **0.800** | +0.002 | 0.946 | **0.949** | +0.003 | 0.024 | **0.023** | +0.001 | 0.952 | **0.960** | +0.008 |
| Gaussian ($\sigma = 0.01, \mu = 0.02$) | 0.661 | **0.666** | +0.005 | 0.875 | **0.898** | +0.023 | **0.038** | 0.039 | -0.001 | **0.549** | 0.524 | -0.025 |
| mean | 0.836 | **0.855** | +0.019 | 0.927 | **0.950** | +0.023 | 0.020 | **0.019** | +0.001 | 0.917 | **0.912** | -0.005 |

## 6 CONCLUSION

We introduced NoiseSDF2NoiseSDF, a framework that recovers clean surfaces from noisy, sparse point clouds in a Noise2Noise denoising approach. At noise levels of 0.01 and 0.02, our method produces cleaner and smoother surfaces, both quantitatively and visually, compared to previous works. In the future, we aim to explore additional applications of the NoiseSDF2NoiseSDF, such as scaling point cloud sizes for more complex geometry or replacing framework components with alternative architectures to improve noise representation and denoising performance.

## 7 ETHICS STATEMENT

Our work, NoiseSDF2NoiseSDF, adheres to the ICLR Code of Ethics. The research does not involve human subjects or animals, and all datasets used—ShapeNet, ABC, Famous, and Real—are publicly available with proper citations. The models used in our experiments are also open-source and distributed under appropriate licenses. We confirm that there are no conflicts of interest related to this work. Additionally, we have ensured that the privacy and security of the data used in our experiments comply with relevant data protection guidelines.

## 8 REPRODUCIBILITY STATEMENT

To ensure the reproducibility of our work, the code script is provided in the supplementary materials, see Appendix A.7. Furthermore, we will publicly release the complete source code repository along with step-by-step instructions to facilitate easy replication of our results.

For comprehensive details on the experimental setup—including the training procedure, evaluation metrics, test sets, computational resources, and hyperparameter configurations—readers are directed to Sections 4.2 and 5.

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

# A    APPENDIX

### APPENDIX A.1    POTENTIAL IMPACT AND APPLICATIONS

The proposed NoiseSDF2NoiseSDF framework enables denoising of signed distance fields directly from noisy supervision, making it applicable in settings where clean ground-truth data is difficult or costly to obtain. This can enhance the robustness of 3D reconstruction in domains such as robotics, augmented reality, cultural preservation, and medical imaging, where input data is often imperfect. By removing the dependency on clean supervision, the method also reduces dataset curation overhead and energy consumption.

Looking forward, the core idea underlying our framework—a Noise2Noise-style supervision for implicit representations—may extend beyond SDFs. We believe it opens a promising direction for training NeRFs or other neural fields from noisy observations, offering a generalizable, reusable solution for learning continuous representations from imperfect data.

### APPENDIX A.2    THE USE OF LARGE LANGUAGE MODELS

Our research does not involve large language models in any component of the core methodology. We only use LLMs to refine our writing, such as performing appropriate language translations, correcting grammatical errors, and helping adjust the phrasing of certain sentences in the paper.

### APPENDIX A.3    METRICS FORMULA

We detail the evaluation metrics adopted in our experiments.

$$\mathbf{CD}(P,Q) = \frac{1}{|P|} \sum_{p \in P} \min_{q \in Q} \|p - q\|_2 \; + \; \frac{1}{|Q|} \sum_{q \in Q} \min_{p \in P} \|q - p\|_2,$$

$$\mathbf{F}_1(\tau) = \frac{2\,\mathrm{precision}(\tau)\,\mathrm{recall}(\tau)}{\mathrm{precision}(\tau) + \mathrm{recall}(\tau)}, \quad \tau = 0.02,$$

$$\mathbf{NC} = \frac{1}{N} \sum_{i=1}^{N} |\langle n_i^{\mathrm{pred}}, n_i^{\mathrm{gt}} \rangle|,$$

$$\mathbf{MNC}(M) = \frac{1}{|E|} \sum_{e=(v_0,v_1) \in E} \left( 1 - \frac{\langle (v_1 - v_0) \times (a_e - v_0), (b_e - v_0) \times (v_1 - v_0) \rangle}{\|(v_1 - v_0) \times (a_e - v_0)\| \, \|(b_e - v_0) \times (v_1 - v_0)\|} \right).$$

Here $P$ and $Q$ are the sets of predicted and ground-truth 3D samples, with $|\cdot|$ denoting cardinality. The Chamfer Distance (**CD**) measures the average nearest-neighbor distance between the two sets in both directions. It is non-negative ($\geq 0$), and lower values indicate better geometric alignment.

The threshold $\tau$ (set to 0.02) in the $\mathbf{F}_1(\tau)$ score defines precision and recall by counting how many nearest-neighbor distances fall below $\tau$. The F1 score, computed as the harmonic mean of precision and recall, ranges from 0 to 1, with higher values indicating better correspondence between predicted and ground-truth points.

In NC, $n_i^{\mathrm{pred}}$ and $n_i^{\mathrm{gt}}$ denote the predicted and ground-truth unit normals at sample $i$ (out of $N$ total samples). The Normal Consistency is the mean absolute dot product between matched normals, ranging from 0 to 1. Higher values imply better alignment of surface orientation.

$\mathrm{MNC}(M)$ denotes Mesh Normal Consistency for a mesh $M$ with edge set $E$. Each edge $e = (v_0, v_1)$ is shared by two faces whose opposite vertices are $a_e$ and $b_e$. The unnormalized face normals are computed via cross products $(v_1 - v_0) \times (a_e - v_0)$ and $(b_e - v_0) \times (v_1 - v_0)$, and their cosine similarity measures the local smoothness across the edge. The value of MNC is averaged over all edges, and typically falls in the range $[0, 2]$, where lower values correspond to smoother and more consistent surface geometry.

APPENDIX A.4    MORE VISUALIZATION

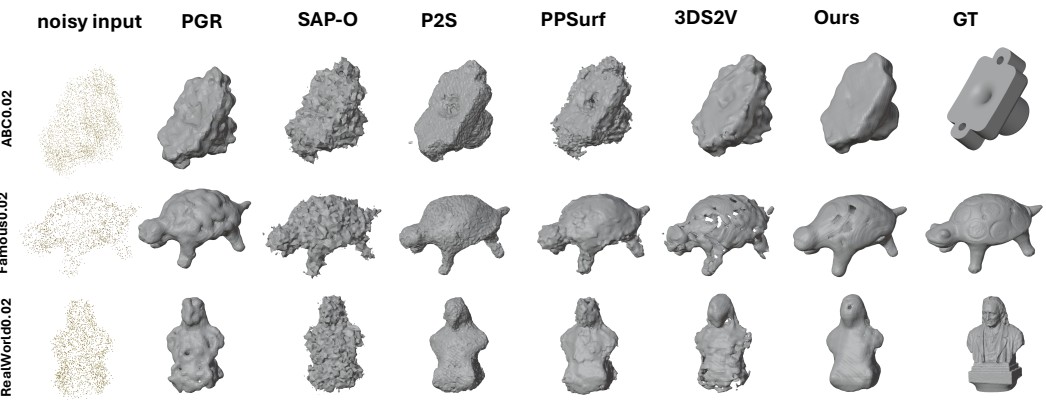

**Figure Appendix A.5:** More results on ABC, Famous and Real under the noise level of 0.02

APPENDIX A.5    MORE QUANTITATIVE RESULTS

APPENDIX A.5.1    DENOISING-THEN-RECONSTRUCTION COMPARISON

To contrast with our end-to-end framework, we evaluate a *two-stage denoising–then–reconstruction* pipeline, in which point cloud denoising and surface reconstruction are performed as separate mod-

**Table Appendix A.9:** Comparison of the SOTA method Neural-Singular-Hessian (NSH) (Wang et al., 2023b), the traditional method Poisson Reconstruction (Poisson), and NoiseSDF2NoiseSDF (Ours) across six noisy test datasets.

| Dataset ($\sigma$) | NC ↑ | | | Mesh NC ↓ | | | Chamfer ↓ | | | F-Score ↑ | | |
|---|---|---|---|---|---|---|---|---|---|---|---|---|
| | NSH | Poisson | Ours | NSH | Poisson | Ours | NSH | Poisson | Ours | NSH | Poisson | Ours |
| ABC (0.01) | 0.845 | 0.804 | **0.865** | 0.026 | 0.074 | **0.024** | 0.016 | 0.058 | **0.014** | 0.933 | 0.771 | **0.938** |
| ABC (0.02) | 0.752 | 0.743 | **0.812** | 0.019 | 0.091 | **0.018** | **0.026** | 0.078 | 0.032 | **0.786** | 0.679 | 0.724 |
| Famous (0.01) | 0.784 | 0.783 | **0.831** | 0.028 | 0.072 | **0.025** | 0.017 | 0.036 | **0.016** | 0.941 | 0.847 | **0.941** |
| Famous (0.02) | 0.706 | 0.728 | **0.767** | 0.035 | 0.090 | **0.024** | **0.027** | 0.049 | 0.034 | **0.760** | 0.757 | 0.726 |
| Real (0.01) | 0.790 | 0.806 | **0.845** | **0.030** | 0.065 | 0.031 | 0.016 | 0.052 | **0.015** | 0.953 | 0.783 | **0.956** |
| Real (0.02) | 0.713 | 0.721 | **0.793** | 0.023 | 0.078 | **0.020** | **0.025** | 0.087 | 0.026 | 0.783 | 0.643 | **0.809** |
| **mean (0.01)** | 0.806 | 0.798 | **0.847** | 0.028 | 0.070 | **0.027** | 0.016 | 0.049 | **0.015** | 0.942 | 0.800 | **0.945** |
| **mean (all)** | 0.765 | 0.764 | **0.819** | 0.027 | 0.078 | **0.024** | **0.021** | 0.060 | 0.023 | **0.859** | 0.747 | 0.849 |

ules. Specifically, we employ IterativePFN (de Silva Edirimuni et al., 2023) for point cloud denoising and use 3DS2V as the subsequent Point2SDF surface reconstruction module. The evaluation is conducted on the ShapeNet Chair category under Gaussian noise of $\sigma = 0.01$, and the quantitative results in Table Appendix A.10 show that our method achieves superior reconstruction quality.

**Table Appendix A.10:** Comparison between a two-stage denoising–then–reconstruction pipeline and NoiseSDF2NoiseSDF (Ours).

| Method | IoU ↑ | NC ↑ | CD-L2 ↓ | F-Score ↑ |
|---|---|---|---|---|
| Denoising-Then-Reconstruction | 0.882 | 0.954 | 0.016 | 0.969 |
| **Ours** | **0.927** | **0.966** | **0.013** | **0.986** |

## APPENDIX A.5.2    MULTIPLE NOISE-TO-NOISE MAPPINGS

We explore whether extending NoiseSDF2NoiseSDF to a multi-target Noise2Noise formulation can further stabilize training. In addition to the standard pairwise (1-to-1) setup, we implement a *1-to-3 Noise2Noise mapping*, where in each training step the Point2SDF network processes three independently corrupted point clouds, and the mean of their predicted SDF values is used as supervision. The experiment follows the ablation settings on the ShapeNet Chair dataset with Gaussian noise of $\sigma = 0.01$. The 1-to-3 configuration indeed produces a slightly more stable training process with lower loss, but it does not improve reconstruction quality: the final IoU, NC, Chamfer, and F-Score are essentially the same as in the 1-to-1 case, as summarized in Table Appendix A.11.

**Table Appendix A.11:** Quantitative comparison between the standard 1-to-1 Noise2Noise formulation and the 1-to-3 multi-target variant.

| Method | IoU ↑ | NC ↑ | Chamfer ↓ | F-Score ↑ |
|---|---|---|---|---|
| 3DS2V | 0.887 | 0.937 | 0.014 | 0.986 |
| 1-to-1 | 0.927 | 0.966 | 0.013 | 0.986 |
| 1-to-3 | 0.923 | 0.967 | 0.013 | 0.986 |

## APPENDIX A.5.3    REALISTIC LiDAR SENSING CONDITIONS

To assess robustness under more realistic sensing conditions, we further evaluate our method using a LiDAR-style noise model. Unlike additive i.i.d. perturbations, real range sensors exhibit structured artifacts such as depth-dependent noise, spatially irregular sampling, dropout, and sporadic outlier returns. These effects introduce anisotropic and highly non-uniform corruption patterns that pose a greater challenge for surface reconstruction.

We consider three corruption levels, with higher levels introducing stronger depth noise, more aggressive dropout, and a larger proportion of outlier returns. The evaluation is conducted on the ShapeNet Chair dataset, where ground-truth meshes are available for quantitative assessment. The corresponding results are summarized in Table Appendix A.12, covering both Chamfer Distance (CD) and F1 under progressively severe LiDAR-style corruptions.

**Table Appendix A.12:** Performance under LiDAR-style structured noise at three corruption levels on the ShapeNet Chair dataset.

| Corruption Level | CD $\downarrow$ | | F1 $\uparrow$ | |
| --- | --- | --- | --- | --- |
| | 3DS2V | Ours | 3DS2V | Ours |
| **Low** | 0.016 | **0.012** | 0.961 | **0.990** |
| **Mid** | 0.031 | **0.023** | 0.772 | **0.878** |
| **High** | 0.059 | **0.052** | 0.518 | **0.601** |

APPENDIX A.6    THEORETICAL JUSTIFICATION

We present a first-order analysis showing that perturbing the closest point with zero-mean Gaussian noise yields a noisy signed distance whose expectation, to first order, approximately matches the clean signed distance at the query point, including on the zero level set.

Let $\phi : \mathbb{R}^d \to \mathbb{R}$ be a signed distance function with zero level set $S = \{x : \phi(x) = 0\}$, satisfying the Eikonal equation $|\nabla\phi(x)| = 1$. For a query point $q$, let $p$ be its closest point on $S$, define the unit normal $n := \nabla\phi(p)$, and write

$$q = p + sn, \quad s = \phi(q).$$

We consider a perturbation of the closest point $p$ by zero-mean Gaussian noise, $\tilde{p} = p + \varepsilon$ with $\varepsilon \sim \mathcal{N}(0, \Sigma_p)$, and study the induced perturbation of the signed distance, $\delta s := \tilde{s} - s$.

**Analysis**: We perform a first-order Taylor expansion around $p$:

$$\phi(x) \approx \phi(p) + \nabla\phi(p)^\top (x - p) = n^\top (x - p),$$

since $\phi(p) = 0$. Thus, near $p$ the signed distance is approximately the projection onto the normal. Approximating the signed distance with respect to the perturbed reference $\tilde{p}$ and treating $n$ as constant to first order, we obtain

$$\tilde{s} \approx n^\top (q - \tilde{p}) = n^\top (q - (p + \varepsilon)) = n^\top (q - p) - n^\top \varepsilon \approx s - n^\top \varepsilon.$$

Hence,

$$\delta s = \tilde{s} - s \approx -n^\top \varepsilon,$$

which is a linear functional of $\varepsilon$. Since $\varepsilon$ is zero-mean Gaussian, we have

$$\mathbb{E}[\delta s] \approx -n^\top \mathbb{E}[\varepsilon] = 0, \quad \text{and therefore} \quad \mathbb{E}[\tilde{s}(q)] \approx s(q).$$

Crucially, this reasoning does *not* invoke a condition of the form $|\varepsilon| \ll |s(q)|$ and remains valid at the zero-level set. If $q \in S$, then $s = \phi(q) = 0$ and $q = p$, so the first-order expression becomes

$$\tilde{s} \approx -n^\top \varepsilon, \quad \delta s \approx -n^\top \varepsilon,$$

with

$$\mathbb{E}[\tilde{s}] \approx 0 = s.$$

Overall, this normal-based analysis provides a first-order unbiasedness guarantee for the noisy SDF both away from and exactly at the zero-level set.

APPENDIX A.7    OTHER SUPPLEMENTARY MATERIALS

To ensure the reproducibility of our work, we provide the code in a zip file, which includes scripts for model training, evaluation, and testing. Additionally, it contains the dataset source and the basic data preprocessing pipeline we used. For further details, please refer to the README.md file.

