# OpenReview forum: "NoiseSDF2NoiseSDF: Learning Clean Neural Fields from Noisy Supervision"
_ICLR.cc/2026/Conference — Submitted to ICLR 2026_

### Official Review · Reviewer_gKoz · 2025-10-24

**Soundness:** 3
**Presentation:** 3
**Contribution:** 2
**Rating:** 6
**Confidence:** 4

**Summary:**

This paper introduces a diffusion-based training framework for 3D shape representations using mesh (SDF) supervision, inspired by the Noise2Noise paradigm.
By extending the idea of pixel-to-pixel correspondences in the 2D image domain to coordinate-to-coordinate correspondences in 3D, the authors propose to learn the probabilistic distribution of SDF values derived from noisy point clouds, showing a reasonable improvement over the baseline model.

**Strengths:**

- Proposes a simple approach to train a point-to-SDF model under the Noise2Noise learning paradigm.

- Provides solid experimental results and comparisons against established baselines.

**Weaknesses:**

- The contribution appears rather incremental. The proposed framework effectively combines 3DShape2VecSet with the Noise2Noise training scheme, without introducing substantial architectural or theoretical novelty.

- In volumetric 3D representations, spatial correspondences are typically defined through voxel, which directly parallels pixel correspondences in 2D. In contrast, SDF values encode continuous geometric distances, not discrete presence. It is unclear why SDFs are selected in the paper.

- While the method works empirically, the theoretical justification for using SDF-based correspondences remains weak. The training process implicitly assumes that coordinate-level SDF fields under noise satisfy the unbiased expectation property (as in Noise2Noise), but this assumption is non-trivial given the nonlinearity and spatial correlation of SDF fields.

**Questions:**

- The injected noise level appears quite small. Does this mean that the perturbation in the 3DShape2VecSet latent space is minimal, and thus the diffusion process only acts as a minor regularization?

- The Noise2Noise framework relies on i.i.d., zero-mean noise assumptions, ensuring that averaging multiple noisy observations converges to the clean signal as introduced in section 3 of this paper. However, SDF values are nonlinear and spatially correlated in 3D space, which breaks this assumption. While the authors attempt to mitigate this issue by injecting noise at the point-cloud level before constructing SDFs, this process can still introduce a nonzero expectation bias. The paper would be much stronger if the authors provided theoretical evidence that the unbiased expectation property, $E\[\hat{s}(q|p_i)] = s(q)$ in $\forall i \in {1, 2, ..., n}$, approximately holds, or that such bias is negligible, under their noise-injection scheme; otherwise, the method appears to simply demonstrate that applying Noise2Noise to SDFs works empirically, without offering clear theoretical justification.

Since these concerns (particularly the justification for using SDFs instead of voxel and the validity of Noise2Noise assumptions under geometric noise) lies at the core of the proposed method, my overall evaluation would strongly depend on how it is addressed.

---

> ### Author Response · Authors · 2025-11-21
> **Authors' Rebuttal (Part 1/2)**
>
> # Q1: Contribution
>
> We appreciate the reviewer’s concern regarding the perceived incrementality of our contribution. Our intention in this work is not to introduce a new network architecture, but rather to address a different and, to our knowledge, unexplored question: can Noise2Noise learning be made to work in the space of 3D neural SDF fields, and thus enable learning clean neural fields from noisy supervision without clean SDF labels? This problem has not yet been investigated and, we believe, opens up a promising direction for future research. To this end, NoiseSDF2NoiseSDF (i) extends the Noise2Noise idea from images to neural SDF fields, (ii) introduces a N2N SDF training pipeline where a denoising neural field is supervised by noisy neural SDFs generated from off-the-shelf point-to-SDF methods, and (iii) empirically demonstrates the effectiveness of the proposed approach and training pipeline. We agree that the architecture is not the primary novelty, and this is a deliberate design choice: by reusing a strong existing backbone, we highlight the contribution of the proposed NoiseSDF2NoiseSDF training scheme and the conceptual extension of Noise2Noise to 3D neural fields.
>
> Motivated by the reviewer’s comment, we have added a theoretical justification clarifying why the proposed Noise2Noise formulation is appropriate in the neural SDF setting, which we hope better conveys the originality and significance of our framework beyond simply “combining 3DShape2VecSet with Noise2Noise.”
>
> # Q2: SDF selected
>
> We appreciate the reviewer’s question about why we choose SDFs rather than voxel-based volumetric grids. While voxel grids do provide a direct 3D analogue to pixels via discrete indices, they make robust, high-fidelity surface reconstruction from noisy point clouds difficult, as dense volumetric grids are memory-intensive and tend to struggle with capturing fine geometric details. Neural SDFs instead define a continuous function, giving us a stable coordinate domain analogous to image pixels but without cubic memory growth, and allowing arbitrary sampling near the surface. At each query coordinate, an SDF assigns a scalar signed distance, providing the same coordinate-based alignment as pixels or voxels.
>
> # Q3: Noise level
>
> The injected noise levels used in our experiments are not small for 3DShape2VecSet, and the resulting perturbations in its latent space are therefore far from minimal. The values we adopt (0.01 and 0.02) follow the standard practice in prior work on noisy point clouds and are known to produce meaningful geometric corruption. Empirically, even the lower noise level already causes a substantial performance drop. As shown in Table 1, applying a noise level of 0.01 decreases the IoU of 3DShape2VecSet from 0.967 on clean point clouds (as reported in its original paper) to 0.884. Under the same noisy conditions, our method raises the IoU from 0.884 to 0.929, indicating that the degradation is significant and that our approach effectively counteracts it. Further evidence comes from Tables 1 and 2: increasing the noise level to 0.02 produces an even larger performance decline in the 3DShape2VecSet baseline, confirming that the perturbations are non-trivial. In addition, the qualitative results in Figure 3 clearly show strong visual degradation under both noise levels and correspondingly strong improvements produced by our method. Therefore, these observations show that the injected noise substantially disrupts the latent representation in 3DShape2VecSet. The denoising process in our method is not a minor regularization effect but a critical component for restoring accurate geometry under realistic and impactful noise levels.

---

> > ### Author Response · Authors · 2025-11-21
> > **Authors' Rebuttal (Part 2/2)**
> >
> > # Q4: Theoretical justification
> >
> > We thank the reviewer for highlighting the need for a theoretical justification of the expectation property. We agree that, due to the nonlinearity of SDF generation, this assumption is not trivial. Below we provide a first-order analysis showing that, under standard small-noise conditions, the noisy SDF can be regarded as an approximately estimator of the clean SDF at each query point.
> >
> > **Objective.** . Show that $\mathbb{E}[\tilde{s}(q)] \approx s(q)$, where $s(q)$ is the clean signed distance function (SDF) and $\tilde{s}(q)$ is the SDF derived from a noisy point cloud.
> >
> > **Analysis.**. Let $S$ denote the clean surface. For a query point $q$, let $p$ be the closest point on $S$. Let $n$ be the surface normal at $p$, i.e., $n = \nabla s(p)$, so that locally
> > $$
> > q - p = s(q)\,n.
> > $$
> > We assume that the point cloud is corrupted by zero-mean, i.i.d.\ Gaussian noise. The closest clean point $p$ is perturbed to
> > $$
> > \tilde{p} = p + \varepsilon,\quad \varepsilon \sim \mathcal{N}(0, \sigma^2 I_3).
> > $$
> > The noisy SDF value $\tilde{s}(q)$ corresponds to the signed distance from $q$ to the noisy surface. Assuming the noise $\varepsilon$ is sufficiently small so that $\tilde{p}$ remains the locally closest point, the distance from $q$ to $\tilde{p}$ is
> > $$
> > \tilde{d}(q) = \|q - \tilde{p}\|
> > = \|(q - p) - \varepsilon\|
> > = \|s(q)n - \varepsilon\|
> > = \sqrt{s(q)^2\|n\|^2 - 2s(q)(n \cdot \varepsilon) + \|\varepsilon\|^2}.
> > $$
> > Since $\|n\|=1$:
> > $$
> > \tilde{d}(q) = \sqrt{s(q)^2 - 2s(q)(n \cdot \varepsilon) + \|\varepsilon\|^2}
> > $$
> > To obtain a first-order approximation, we assume the noise magnitude is small relative to the distance $(\|\varepsilon\| \ll |s(q)|)$ and neglect the second-order term $\|\varepsilon\|^2$. Using the first-order Taylor expansion, we obtain
> > $$
> > \tilde{d}(q) \approx \sqrt{s(q)^2 - 2s(q)(n \cdot \varepsilon)}
> > \approx |s(q)| - \frac{s(q)(n \cdot \varepsilon)}{|s(q)|}
> > = |s(q)| - \operatorname{sign}(s(q))(n \cdot \varepsilon).
> > $$
> > Returning to the signed distance (where $\tilde{s}(q)$ shares the sign of $s(q)$), we obtain the linear approximation
> > $$
> > \tilde{s}(q) = \operatorname{sign}(s(q))\, \tilde{d}(q) \approx s(q) - n \cdot \varepsilon.
> > $$
> > We now take the expectation of the noisy SDF:
> > $$
> > \mathbb{E}[\tilde{s}(q)] \approx \mathbb{E}[s(q) - n \cdot \varepsilon].
> > $$
> > Since $s(q)$ and $n$ are deterministic properties of the clean surface, and the noise is zero-mean with $\mathbb{E}[\varepsilon] = 0$, we have
> > $$
> > \mathbb{E}[\tilde{s}(q)] \approx s(q) - n \cdot \mathbb{E}[\varepsilon] = s(q).
> > $$

---

> > > ### Comment · Reviewer_gKoz · 2025-11-22
> > >
> > > Thank you for the author's effort to provide a theoretical justification for the unbiased expectation. However, it should be pointed out that the presented derivation fundamentally relies on the assumption
> > >
> > > $|\varepsilon| \ll |s(q)|$
> > >
> > > which is required for the first-order Taylor approximation and for ignoring the $|\varepsilon|^2$ term. This “small-noise relative to SDF magnitude” assumption does not generally hold in the regions that are most critical for surface reconstruction.
> > >
> > > In particular, for points near the underlying surface where $s(q) \rightarrow 0$, the condition $|\varepsilon| \ll |s(q)|$ cannot be satisfied for any non-zero noise level. Therefore, the expansion
> > >
> > > $\tilde d(q) \approx |s(q)| - \mathrm{sign}(s(q)) (n\cdot\varepsilon)$
> > >
> > > would not hold in the zero-level set.
> > >
> > > While the derivation may reasonably approximate the behavior of $\tilde s(q)$ for points sufficiently far from the surface (i.e., where $|s(q)| \gg |\varepsilon|$), it does not provide a theoretical guarantee for the zero-level set from where the reconstructed surface is extracted.
> > >
> > > The proposed analysis does not fully resolve my concern regarding the existence of non-negligible expectation bias under practical noise regimes.
> > >
> > > Unless the authors can offer an alternative theoretical argument or a clearer justification that holds at the zero-level set, it would be difficult to regard the proposed Noise2Noise formulation for SDFs as theoretically valid. Without such clarification or further analysis, a negative recommendation would have to be maintained.

---

> > > > ### Author Response · Authors · 2025-11-25
> > > > **Authors’ Revised Theoretical Justification**
> > > >
> > > > We thank the reviewer for the detailed feedback and agree that our previous derivation, which relied on a condition of the form $|\varepsilon| \ll |s(q)|$, did not adequately address the behavior at the zero-level set, since this ``small noise relative to SDF magnitude'' assumption breaks down as $s(q) \to 0$. The earlier norm-based analysis expressed the SDF as a scalar magnitude with a sign correction, introducing an explicit dependence on the ratio $|\varepsilon| / |s(q)|$ and becoming ill-conditioned near the surface.  We therefore replace it with a normal-based analysis, in which the SDF is modeled along the surface normal as $n^\top (q - p)$. Under this analysis, noise enters linearly as $n^\top \varepsilon$, independent of $|s(q)|$, and the first-order behavior remains well defined even when $s(q) = 0$.
> > > >
> > > > Let $\phi : \mathbb{R}^d \to \mathbb{R}$ be a signed distance function with zero level set $S = \{ x : \phi(x) = 0 \}$, satisfying the Eikonal equation $|\nabla \phi(x)| = 1$. For a query point $q$, let $p$ be its closest point on $S$, define the unit normal $n := \nabla \phi(p)$, and write
> > > > $$
> > > > q = p + s n, \quad s = \phi(q).
> > > > $$
> > > > We consider a perturbation of the closest point $p$ by zero-mean Gaussian noise, $\tilde p = p + \varepsilon$ with $\varepsilon \sim \mathcal{N}(0, \Sigma_p)$, and study the induced perturbation of the signed distance, $\delta s := \tilde s - s$.
> > > > We perform a first-order Taylor expansion around $p$:
> > > > $$
> > > > \phi(x) \approx \phi(p) + \nabla \phi(p)^\top (x - p) = n^\top (x - p),
> > > > $$
> > > > since $\phi(p) = 0$. Thus, near $p$ the signed distance is approximately the projection onto the normal. Approximating the signed distance with respect to the perturbed reference $\tilde p$ and treating $n$ as constant to first order, we obtain
> > > > $$
> > > > \tilde s \approx n^\top (q - \tilde p) = n^\top (q - (p + \varepsilon)) = n^\top (q - p) - n^\top \varepsilon \approx s - n^\top \varepsilon.
> > > > $$
> > > > Hence,
> > > > $$
> > > > \delta s = \tilde s - s \approx -n^\top \varepsilon,
> > > > $$
> > > > which is a linear functional of $\varepsilon$. Since $\varepsilon$ is zero-mean Gaussian, we have
> > > > $$
> > > > \mathbb{E}[\delta s] \approx -n^\top \mathbb{E}[\varepsilon] = 0, \quad \text{and therefore} \quad \mathbb{E}[\tilde s(q)] \approx s(q).
> > > > $$
> > > > Crucially, this reasoning does \emph{not} invoke a condition of the form $|\varepsilon| \ll |s(q)|$ and remains valid at the zero-level set. If $q \in S$, then $s = \phi(q) = 0$ and $q = p$, so the first-order expression becomes
> > > > $$
> > > > \tilde s \approx -n^\top \varepsilon, \quad \delta s \approx -n^\top \varepsilon,
> > > > $$
> > > > with
> > > > $$
> > > > \mathbb{E}[\tilde s] \approx 0 = s.
> > > > $$
> > > > Overall, this normal-based analysis provides a first-order unbiasedness guarantee for the noisy SDF both away from and exactly at the zero-level set. We hope this resolves the reviewer’s concern regarding the zero-level set and clarifies the theoretical justification of using a Noise2Noise formulation for SDFs under practical noise regimes.

---

> > > > > ### Comment · Reviewer_gKoz · 2025-11-25
> > > > >
> > > > > While the normal-based analysis proposed by the authors yields an unbiased expectation under the linear approximation
> > > > >
> > > > > $\phi(q) \approx n^{\top}(q - p),$
> > > > >
> > > > > this approximation is not equivalent to the actual SDF.
> > > > >
> > > > > The true noisy SDF under a perturbation of the closest point $ p $ by Gaussian noise $ \varepsilon $ is given by
> > > > >
> > > > > $
> > > > > \tilde{s}(q)
> > > > > = \| q - (p + \varepsilon) \|
> > > > > = \sqrt{s(q)^2 - 2 s(q) (n \cdot \varepsilon) + \|\varepsilon\|^2\},$
> > > > >
> > > > > as discussed in the initial comment of the author. However, the normal-based derivation provided by the authors replaces the true SDF with the
> > > > >
> > > > > $\tilde{s}(q) \approx n^{\top}(q - (p + \varepsilon)) \approx s(q) - n^{\top}\varepsilon,$
> > > > >
> > > > > which removes all nonlinear terms and implicitly assumes $\| \varepsilon \| \ll \| s(q) \|$.
> > > > >
> > > > > As a result, the unbiased expectation obtained under this linearized model does not correspond to the expectation of the actual SDF under noise, particularly near the zero-level set where $s(q)=0$ and the nonlinear bias is unavoidable.
> > > > >
> > > > > Therefore, the normal-based analysis does not resolve the fundamental expectation bias inherent to the signed distance formulation.

---

> > > > > > ### Author Response · Authors · 2025-11-26
> > > > > >
> > > > > > We thank the reviewer for the additional clarifications and respond to each comment in turn. First, we would like to clarify that both the norm-based model and the normal-based model are widely used \emph{approximate local models}, derived under different geometric assumptions. In particular, norm-based models rely on the (unsigned) point-to-point distance to a single surface sample, whereas the normal-based model treats the SDF as the signed distance to a locally planar surface patch.
> > > > > >
> > > > > > * “... under the linear approximation $\phi(q) \approx n^\top(q - p)$ this approximation is not equivalent to the actual SDF.”
> > > > > >
> > > > > > We understand that the reviewer's concern regarding the initial analysis, and therefore we provided the second attempt of normal-based analysis based on
> > > > > > $$
> > > > > > s(q) = \phi(q) \approx n^\top(q-p).
> > > > > > $$
> > > > > > We agree that the linear approximation is not identical to the exact SDF. However, this is the standard local planar normal-based model for the signed distance function, which is generally regarded as the canonical model and has been widely applied in many highly related literature, including Neural-Pull (Ma et al., 2020), Implicit Filtering-Net (Li et al., 2024), and [1,2], etc. These works have been published in top-tier peer-reviewed venues and have already been followed by many researchers in the community.
> > > > > >
> > > > > > 1. B Ma, YS Liu, Z Han. Reconstructing surfaces for sparse point clouds with on-surface priors. CVPR 2022
> > > > > > 2. C Chen, YS Liu, Z Han. GridPull: Towards scalability in learning implicit representations from 3d point clouds. ICCV 2023
> > > > > >
> > > > > >
> > > > > > * “The true noisy SDF ... is given by $ \tilde s(q) = \|q - (p+\varepsilon)\|
> > > > > > = \sqrt{s(q)^2 - 2s(q)(n\cdot\varepsilon) + \|\varepsilon\|^2}  $ as discussed in the initial comment of the author”
> > > > > >
> > > > > > We would like to clarify that in our initial comment, we did not claim “$ \tilde s(q) = \|q - (p+\varepsilon)\|
> > > > > > = \sqrt{s(q)^2 - 2s(q)(n\cdot\varepsilon) + \|\varepsilon\|^2}  $ ” as the "true noisy SDF". Instead, we explicitly stated that $\sqrt{s(q)^2 - 2s(q)(n\cdot\varepsilon) + \|\varepsilon\|^2} $  is the distance from $q$ to $\tilde{p}$.
> > > > > >
> > > > > >
> > > > > > * “However, the normal-based derivation provided by the authors replaces the true SDF with $\tilde s(q) \approx n^\top(q-(p+\varepsilon)) \approx s(q) - n^\top\varepsilon,$ which removes nonlinear terms and implicitly assumes $|\varepsilon| \ll |s(q)|$.”
> > > > > >
> > > > > > We would like to clarify that there is no “true SDF” being replaced here. Our initial analysis relied on the norm-based model. Our revised analysis adopts the normal-based SDF model, which is the standard first-order SDF model used in prior work and does not involve square root or absolute value. Under this normal-based planar model, we get
> > > > > > $$
> > > > > > \tilde s(q) \approx s(q) - n^\top\varepsilon.
> > > > > > $$
> > > > > > In this setting, no nonlinear terms are dropped, and the derivation does not involve the magnitude of $s(q)$. Thus, the model does not impose or rely on an assumption of $|\varepsilon| \ll |s(q)|$.
> > > > > > Moreover, for zero-mean perturbations, we have
> > > > > > $$
> > > > > > \mathbb{E}[\tilde s(q)] = s(q)
> > > > > > $$
> > > > > > holds both far from the surface and exactly at the zero level set.
> > > > > >
> > > > > >
> > > > > > * “As a result, the unbiased expectation obtained under this linearized model does not correspond to the expectation of the actual SDF under noise, particularly near the zero-level set where $s(q)=0$
> > > > > >  and the nonlinear bias is unavoidable.”
> > > > > >
> > > > > > Under a point-to-point norm-based model, a nonlinear bias near $s(q)=0$ is indeed unavoidable. However, under the standard normal-based SDF model, the noisy signed distance satisfies $\tilde s(q) = s(q) - n^\top\varepsilon$, so zero-mean perturbations lead to $\mathbb{E}[\tilde s(q)] = s(q)$, even at the zero level set.
> > > > > >
> > > > > >
> > > > > > * “Therefore, the normal-based analysis does not resolve the fundamental expectation bias inherent to the signed distance formulation.”
> > > > > >
> > > > > > Within the revised analysis based on normal-based model of the SDF, the first-order expectation bias is fully resolved: perturbations shift the signed distance linearly in $n^\top\varepsilon$, whose expectation is zero under zero-mean noise.

---

> > > > > > > ### Comment · Reviewer_gKoz · 2025-11-27
> > > > > > >
> > > > > > > Thank you to the authors for clarifying the parts I initially misunderstood. After reviewing the materials referenced in the comments and reading related works, Noise2Noise framework can indeed be applied without issue when the underlying representation is a normal-based model rather than a strict Euclidean SDF. In that context, the formulation behaves well and the noise propagation becomes consistent with the assumptions of Noise2Noise.
> > > > > > >
> > > > > > > However, in its current form, the contribution of this paper appears limited. The method primarily combines Noise2Noise with one of the existing normal-based models for implicit SDF representation, without introducing a new theoretical insight, a novel algorithmic component, or a clearly distinguished ability beyond prior work. As a result, the paper reads more as an engineering application of two known ideas rather than a work with substantial conceptual advancement.
> > > > > > >
> > > > > > > Given these considerations, I will change my decision as a weak accept.

---

> > > > > > > > ### Author Response · Authors · 2025-12-04
> > > > > > > >
> > > > > > > > We thank Reviewer gKoz for engaging in a detailed theoretical discussion and, after reviewing the materials referenced in the comments and related works, for acknowledging that our normal-based model’s theoretical justification “can indeed be applied without issue.” Based on this, we believe that we have addressed the concern regarding “theoretical insight.” We clarify that our main contribution is a new training paradigm for learning clean neural fields from noisy supervision. We emphasize that, to the best of our knowledge, this is the first work to formulate and validate a Noise2Noise training hypothesis in neural fields. Therefore, our contribution goes beyond an engineering application and provides a substantial conceptual advancement. This perspective is reflected in other reviews, which describe the key idea as “novel and impactful” (wKfe) and “conceptually novel and elegant” (ZSTJ). We hope our work will further motivate research on learning neural fields from noisy supervision.

---

### Official Review · Reviewer_ZSTJ · 2025-10-27

**Soundness:** 3
**Presentation:** 3
**Contribution:** 3
**Rating:** 6
**Confidence:** 4

**Summary:**

This paper proposes NoiseSDF2NoiseSDF, a novel self-supervised learning framework that extends the Noise2Noise paradigm to 3D neural fields for surface reconstruction from noisy point clouds. The method trains a neural SDF network to predict clean signed distance functions by minimizing the MSE between pairs of noisy SDFs generated from independently corrupted point clouds of the same shape. The paper is clearly written, well-motivated, and presents extensive evaluations validating the idea that clean neural fields can be learned from noisy supervision.

**Strengths:**

The core idea—applying Noise2Noise principles to neural SDF learning—is elegant, conceptually novel, and supported by solid experimental results.

The method is practical: it does not require clean supervision, generalizes across datasets, and achieves high-quality reconstructions even under heavy noise.

**Weaknesses:**

The experimental comparison, while broad, still lacks evaluation against the most recent state-of-the-art neural reconstruction approaches, such as Neural-Singular-Hessian, which would help establish a clearer performance frontier.

Traditional geometry-based methods such as iPSR, PGR, or Poisson Reconstruction should be compared more explicitly to show how the proposed method performs against classical baselines under varying noise levels.

The robustness analysis could be expanded: it would be valuable to include ablation studies on different input point cloud properties, such as varying point density, non-uniform distributions, and missing points.

**Questions:**

Would it be feasible to train NoiseSDF2NoiseSDF jointly across multiple noisy-to-noisy mappings (beyond pairwise), similar to ensemble self-consistency methods, to further stabilize learning?

Is it possible to compare with some point cloud denoising methods? And reconstruct it by other methods after denoising? (Just a suggestion)

---

> ### Author Response · Authors · 2025-11-21
> **Authors' Rebuttal (Part 1/2)**
>
> # Q1: Neural-Singular-Hessian and Poisson comparison
>
> Following the reviewer’s suggestion, we compare against two additional representative baselines: the overfitting-based method Neural-Singular-Hessian (NSH) and the traditional Poisson Reconstruction. Using the same experimental setup as in our main paper, we evaluate these methods on the ABC, Famous, and Real datasets under two noise levels. Neural-Singular-Hessian, as a recent state-of-the-art overfitting method, achieves competitive performance on Chamfer Distance and F1 Score. However, due to its overfitting nature, the network progressively fits the noise during optimization, leading to inferior results on NC and Mesh NC. As a traditional non-learning method, Poisson Reconstruction suffers significantly from noise in the input. As shown in the table, NoiseSDF2NoiseSDF consistently demonstrates good denoising capability relative to these baselines.
>
> | **Dataset (σ)**   | NSH      | Poisson | Ours      | NSH           | Poisson | Ours      | NSH           | Poisson | Ours      | NSH           | Poisson | Ours      |
> | ----------------- | -------- | ------- | --------- | ------------- | ------- | --------- | ------------- | ------- | --------- | ------------- | ------- | --------- |
> |                   | **NC ↑** |         |           | **Mesh NC ↓** |         |           | **Chamfer ↓** |         |           | **F-Score ↑** |         |           |
> | **ABC (0.01)**    | 0.845    | 0.804   | **0.865** | 0.026         | 0.074   | **0.024** | 0.016         | 0.058   | **0.014** | 0.933         | 0.771   | **0.938** |
> | **ABC (0.02)**    | 0.752    | 0.743   | **0.812** | 0.019         | 0.091   | **0.018** | **0.026**     | 0.078   | 0.032     | **0.786**     | 0.679   | 0.724     |
> | **Famous (0.01)** | 0.784    | 0.783   | **0.831** | 0.028         | 0.072   | **0.025** | 0.017         | 0.036   | **0.016** | 0.941         | 0.847   | **0.941** |
> | **Famous (0.02)** | 0.706    | 0.728   | **0.767** | 0.035         | 0.090   | **0.024** | **0.027**     | 0.049   | 0.034     | **0.76**      | 0.757   | 0.726     |
> | **Real (0.01)**   | 0.790    | 0.806   | **0.845** | **0.030**     | 0.065   | 0.031     | 0.016         | 0.052   | **0.015** | 0.953         | 0.783   | **0.956** |
> | **Real (0.02)**   | 0.713    | 0.721   | **0.793** | 0.023         | 0.078   | **0.020** | **0.025**     | 0.087   | 0.026     | 0.783         | 0.643   | **0.809** |
> | **mean (0.01)**   | 0.806    | 0.798   | **0.847** | 0.028         | 0.070   | **0.027** | 0.016         | 0.049   | **0.015** | 0.942         | 0.800   | **0.945** |
> | **mean (all)**    | 0.765    | 0.764   | **0.819** | 0.027         | 0.078   | **0.024** | **0.021**     | 0.060   | 0.023     | **0.859**     | 0.747   | 0.849     |
>
> #  Q2: Robustness analysis
>
> We appreciate the reviewer’s suggestion regarding robustness to varying point densities, non-uniform sampling, and missing points. To further investigate these aspects during the rebuttal stage, we conducted an additional LiDAR scan-noise experiment that exhibits these properties. Because each simulated LiDAR scan is generated via perspective projection and z-buffering, the resulting point clouds inherently contain spatially varying densities—denser around nearly tangent surfaces and sparser at grazing angles. In addition, the depth-dependent dropout mechanism introduces structured missing regions rather than uniform random deletions. These characteristics provide a realistic and diverse set of input conditions for stress-testing the model’s robustness.
>
> Our evaluation includes three corruption levels, where higher levels correspond to stronger depth noise, more aggressive dropout, and a larger proportion of outlier returns. The model maintains stable performance across these settings, indicating that it generalizes well to non-uniform distributions and incomplete observations. All metrics are computed against the ground-truth Chair mesh.
>
> | Corrupted Level | **CD (↓)** |        | **F1 (↑)** |        |
> |-----------------|------------|--------|------------|--------|
> |                 | 3DS2V      | Ours   | 3DS2V      | Ours   |
> | **Low**         | 0.016      | 0.012  | 0.961      | 0.990  |
> | **Mid**         | 0.031      | 0.023  | 0.772      | 0.878  |
> | **High**        | 0.059      | 0.052  | 0.518      | 0.601  |

---

> ### Author Response · Authors · 2025-11-21
> **Authors' Rebuttal (Part 2/2)**
>
> # Q3: Multiple noisy-to-noisy mappings
>
> We thank the reviewer for this suggestion. We conducted experiments to test whether a multi-target noise2noise mapping can further stabilize NoiseSDF2NoiseSDF training. We extended the original pairwise (1-to-1) setup to a 1-to-3 mapping: in each training step, the Point2SDF network processes three independently corrupted point clouds, and their mean predicted SDF is used as supervision. We followed the ablation settings (Chair dataset, Gaussian noise $\sigma = 0.01$). The 1-to-3 configuration indeed produces a slightly more stable training process with lower loss, but it does not improve reconstruction quality: the final IoU, NC, Chamfer, and F-Score are essentially the same as in the 1-to-1 case, as summarized below.
>
> | Method | IoU   | NC    | Chamfer | F-Score |
> |--------|-------|-------|---------|---------|
> | 3DS2V  | 0.887 | 0.937 | 0.014   | 0.986   |
> | 1-to-1 | 0.927 | 0.966 | 0.013   | 0.986   |
> | 1-to-3 | 0.923 | 0.967 | 0.013   | 0.986   |
>
> # Q4: Denoising-Then-Reconstruction comparison
>
> We thank the reviewer for this constructive suggestion. In response, we conducted additional experiments to evaluate the proposed sequential “denoising → reconstruction” pipeline. Specifically, we used IterativePFN (CVPR 2023) for point cloud denoising and 3DS2V as the subsequent Point2SDF surface reconstruction module. We performed this evaluation on the ShapeNet Chair category. The results show that our method achieves superior reconstruction quality.
>
> | Method| IoU ↑ | NC ↑   | CD-L2 ↓ | F-Score ↑ |
> |-----------------------------|-------|--------|---------|-----------|
> | Denoising-Then-Reconstruction | 0.882 | 0.954 | 0.016   | 0.969     |
> | **Ours**                    | **0.927** | **0.966** | **0.013** | **0.986** |

---

### Official Review · Reviewer_wKfe · 2025-10-31

**Soundness:** 3
**Presentation:** 3
**Contribution:** 3
**Rating:** 6
**Confidence:** 4

**Summary:**

This paper introduces NoiseSDF2NoiseSDF, a method that extends the Noise2Noise (N2N) paradigm from 2D image denoising to 3D neural fields. The key idea is to learn a clean neural Signed Distance Function (SDF) from noisy point cloud data by using noisy supervision, minimizing the mean squared error (MSE) loss between them. The authors validate their approach on several benchmarks (ShapeNet, ABC, Famous, Real), demonstrating improvements in surface reconstruction quality over the baseline (3DS2V) and competitive performance against state-of-the-art methods.

**Strengths:**

1. The paper presents a original idea. While N2N is well-established in 2D, its direct application to 3D is difficult due to the unstructured nature of point clouds. The key insight—that neural SDFs provide a structured, continuous representation analogous to pixel grids, thereby enabling the use of simple MSE loss—is novel and impactful.
2. The paper is generally well-written, with a clear pipeline description and visualizations. Figures 1 and 2 effectively illustrate the core concept and framework.
3. Experiments cover multiple datasets and noise levels, and comparisons include relevant baselines and state-of-the-art methods. The ablation studies reinforce the validity of the core claims. The qualitative and quantitative results show the effectiveness of the method, which can generate smoother and more accurate surfaces.

**Weaknesses:**

1. A potential limitation is the method's reliance on a pre-trained Point2SDF network (e.g., 3DS2V) to generate the noisy targets. The paper shows generalizability by swapping in 3DILG, but an analysis of how sensitive the performance is to the quality of this frozen network would be valuable. What happens if the Point2SDF provider is poorly trained and less accurate, trained on a very different domain or trained with noisy data to improve noise roubustness?
2. The method is evaluated primarily on additive Gaussian noise and a few other symmetric, zero-mean noise types. Real-world sensor noise can be more complex, potentially containing non-zero mean biases, outliers, structured artifacts, or incompletion due to obstruction. How does the method perform with real-world noise (e.g., from LiDAR or RGB-D sensors)?
3. Suggest adding a comparison between end-to-end SDF denoising and separate sequential denoising steps. For example, first use point cloud denoising methods (such as TotalDenoising) to denoise the input noisy point cloud, and then feed it into a standard Point2SDF network.

**Questions:**

Refer to Weaknesses.

---

> ### Author Response · Authors · 2025-11-21
> **Authors' Rebuttal**
>
> # Q1: Point2SDF provider
>
> We thank the reviewer for highlighting the role of the pre-trained Point2SDF network in our framework. Our method is explicitly designed as a neural-field Noise2Noise scheme: a frozen Point2SDF model serves as a noisy SDF “camera” that provides corrupted SDF observations, and our network learns a denoised SDF representation by training on these noisy SDF measurements. In the current paper, we evaluate the impact of this component along three aspects:
>
> 1. Different Point2SDF providers. We instantiate NoiseSDF2NoiseSDF with two distinct Point2SDF providers, 3DS2V and 3DILG, and consistently observe that our method improves over the raw provider outputs. This shows that the network is not simply copying the provider, but effectively denoising and refining its predictions. We also observe that a weaker provider leads to weaker absolute performance for both denoising backbones, which is expected since the provider defines the noisy supervision signal, but our method still yields gains over the provider itself.
>
> 2. Different noise levels. We evaluate two noise levels (Gaussian noise with $\sigma$ = 0.01 and $\sigma$ = 0.02). At the higher noise level ($\sigma$ = 0.02), the performance of 3DS2V as the Point2SDF provider drops significantly, effectively behaving as a less accurate SDF estimator. Even in this setting, our training still improves upon the degraded provider, demonstrating that NoiseSDF2NoiseSDF can recover part of the lost quality and is reasonably robust to reduced provider accuracy.
>
> 3. Different domains. In our experiments on the ABC, Famous, and Real, we use a Point2SDF provider trained on ShapeNet, which introduces a domain gap between training and testing distributions. Despite this mismatch, our method achieves strong performance across these datasets, indicating that NoiseSDF2NoiseSDF can effectively adapt to new domains even when the provider is trained on a different dataset.
>
> Overall, we agree that the quality of the Point2SDF provider influences the attainable upper bound and may represent a potential limitation of our method. However, our experiments, across multiple providers, noise levels, and datasets with domain gaps, consistently show that NoiseSDF2NoiseSDF improves over the provider itself and over competing methods.
>
> # Q2: Real-world sensor noise
>
> We appreciate the reviewer’s suggestion to evaluate our method under real-world sensor noise. To this end, we further evaluate it using a LiDAR simulator that mimics a rotating range sensor by performing perspective projection, applying a z-buffer to keep the closest return per ray, and injecting depth-dependent range noise, random dropout, and occasional outlier points. This setup approximates several common characteristics of real LiDAR scans while avoiding the need for paired device-level ground truth.
>
> Our evaluation includes three corruption levels, where higher levels correspond to stronger depth noise, more aggressive dropout, and a larger proportion of outlier returns. Experimental results on the Chair dataset (where the ground-truth mesh is available for quantitative evaluation) show that our model maintains strong robustness under these more realistic corruption patterns. Compared with standard additive noise, LiDAR-style structured noise introduces greater spatial irregularity, yet the NoiseSDF2NoiseSDF training remains stable and consistently improves surface reconstruction. While a full evaluation on real LiDAR hardware is an important future direction, these results suggest that the proposed approach generalizes well beyond symmetric, zero-mean synthetic noise.
>
> | Corrupted Level | **CD (↓)** |        | **F1 (↑)** |        |
> |-----------------|------------|--------|------------|--------|
> |                 | 3DS2V      | Ours   | 3DS2V      | Ours   |
> | **Low**         | 0.016      | 0.012  | 0.961      | 0.990  |
> | **Mid**         | 0.031      | 0.023  | 0.772      | 0.878  |
> | **High**        | 0.059      | 0.052  | 0.518      | 0.601  |
>
>
> # Q3: Denoising-Then-Reconstruction comparison
>
> We thank the reviewer for this constructive suggestion. In response, we conducted additional experiments to evaluate the proposed sequential “denoising → reconstruction” pipeline. Specifically, we used IterativePFN (CVPR 2023) for point cloud denoising, which is more recent and stronger denoising method than earlier approaches such as TotalDenoising (ICCV 2019), and 3DS2V as the subsequent Point2SDF surface reconstruction module. We performed this evaluation on the ShapeNet Chair category. The results show that our method achieves superior reconstruction quality.
>
> | Method| IoU ↑ | NC ↑   | CD-L2 ↓ | F-Score ↑ |
> |-----------------------------|-------|--------|---------|-----------|
> | Denoising-Then-Reconstruction | 0.882 | 0.954 | 0.016   | 0.969     |
> | **Ours**                    | **0.927** | **0.966** | **0.013** | **0.986** |

---

### Official Review · Reviewer_aqUU · 2025-10-31

**Soundness:** 3
**Presentation:** 3
**Contribution:** 1
**Rating:** 2
**Confidence:** 3

**Summary:**

This paper proposes an algorithm for learning Signed Distance Function (SDF) from the noisy pointclouds. The main observation or hypothesis is that the mean of the multiple observations having noise may converge into the clean observation. Following this assumption, the method leverages the neural networks and minimize the consistency loss between the SDF predictions from the two noisy observations.

**Strengths:**

The paper is well written and understandable. The authors smoothly transits the context from the theoretical background from Noise2Noise to the application into the neural fields estimation.

Regarding the hypothesis, '(line 164) ... whether clean neural fields can be effectively learned by observing their noisy counterparts.', it looks reasonable and I personally agree with such an observation. This paper fully focus on proving this assumption and the experiments are well constructed.

Nonetheless, I personally think that the problem definition itself is not novel.

**Weaknesses:**

__W1. Weak problem setup__
In line 175, this paper is designed to _investigate_ whether Noise2Noise concept is also applicable to learning clean neural fields. So, in the equation 5, the loss is to reduce the predicted SDF values from the two noisy observations.

Unlike previous studies that learn SDF from the noisy data, this paper directly and explicitly compute the SDF values from the two noisy observation pairs. I admit that this paper _can_ be the first paper to prove the hypothesis, but it is somehow not concise in my opinion.

Commonly, given noisy points from one observation, the previous work tries to minimize the loss to train the network parameters and learn SDF. Across the multiple training batches, the SDF predictions are obtained from the optimized network parameters and finally converge into the _relatively_ clean surface from the noisy inputs. The difference between such an approach and this submission is that whether to `explicitly` compute the noisy input pairs or not. While the authors propose to compute the loss by explicitly comparing the noisy pairs, but the previous studies `indirectly` do so through different training batches at different training step. The network parameters are shared and optimized to minimize the loss.

Based on this observation, I think that the previous studies _assume_ that Noise2Noisy concept is presumably applicable to the neural field representation, even without the explicit comparison with the two noisy observations at the same time, as suggested by the authors.

So, I am not sure whether the 'explicit' comparison with the multiple noisy observations is needed. Previous studies are dealing with more difficult cases without having such an 'explicit' two noisy pairs.

I hope that the authors resolve such a concern.

__W2 real-world experiments__
My personal preference is that the authors should have tested the hypothesis in the real-world scenarios. For example, the LiDAR points can be posed in the similar problem definition. Depending on the timestamp, the same 3D scenes are differently measured and recorded in each scan. Such measurements are similar to the authors setup that uses multiple and explicit noisy pairs.

**Questions:**

Overall paper is highly well written. I enjoyed reading this paper. However, the problem definition that this paper sets is not something new as stated in __W1__.

Learning clean surfaces from multiple noisy observations are already dealt with the previous studies. The clear difference is whether to have 'explicit' two noisy observations from the same 3D data (this submission), or not. As described in __W1__, I believe that this paper dealt with relatively simple and easy problem compared to recent studies that are trained to predict clean SDF from __single__ and noisy pointclouds without having __multiple__ noisy pairs.

I hope to see the authors' responses in the rebuttals.

---

> ### Author Response · Authors · 2025-11-21
> **Authors' Rebuttal**
>
> # Q1: Weak problem setup
>
> We thank the Reviewer for the detailed comments. We respond here to the core concern that our problem setup is “weak” and that explicit comparison between noisy observations may not be necessary.
>
> First, we clarify the **setting and novelty**. Our method is a **data-driven** neural SDF model that is trained **without access to clean SDF supervision**. Both sides of our training loss are *noisy neural SDF*:
> (i) a learnable SDF predictor that maps noisy point clouds to SDF values, and
> (ii) an independently constructed noisy SDF obtained from an off-the-shelf point-to-SDF method.
> This is not just “two noisy observations” in a vague sense, but a deliberate construction that enables a Noise2Noise (N2N)–style objective in **SDF space**.
>
> In contrast, prior work that learn SDF from the noisy data falls mainly into two categories:
>
> 1. **Per-shape overfitting-based methods.** These methods optimize a separate network per shape given a noisy point cloud, with losses based on, such as, distance constraints, gradient regularization, or EMD. For any given sample, there is typically **only one supervision value** that is repeatedly used during optimization. Seeing different training batches over iterations typically does *not* produce multiple independent noisy SDF labels per coordinate. While there is prior work (N2NM) that uses multiple noisy observations to fit a single shape, it formulates the N2N in the unstructured point-cloud domain with an EMD loss rather than in the SDF domain and still in a per-shape, overfitting paradigm. Therefore, these per-shape overfitting-base methods do not implement an N2N objective at the same query coordinates within the SDF field.
>
> 2. **Data-driven methods.** In this setting, a single network is trained over a large collection of shapes and can map noisy point clouds to clean neural SDFs without further optimization at inference. Existing data-driven methods (e.g., POCO, NKSR) are generally trained using **clean SDFs supervision** computed from high-quality meshes or dense point clouds, thus follow a Noise2Clean training paradigm with clean labels.
>
> With this in mind, we respectfully disagree with the reviewer’s statement that the only difference between prior work and our method is whether we “explicitly compute the noisy input pairs.” The key differences are:
> * Prior data-driven methods assume **clean SDF supervision**; we do not.
> * Prior per-shape overfitting-based methods either use a **single label per point** or formulate N2N objectives only in the unstructured point-cloud domain with an EMD loss; we explicitly construct and exploit such noisy supervision in neural SDF space.
> * Our loss is a **neural-field-to-neural-field MSE** between noisy SDFs, which directly instantiates an N2N-type estimator of the clean SDF, a formulation that, to our knowledge, prior work neither proposes nor analyzes.
>
> Our method tackles a more challenge setting in which clean SDFs are **not available** and only noisy SDF estimates can be used as supervision. In this problem, an N2N formulation is not optional but a much-needed mechanism to enable effective learning from noisy supervision alone for data-driven methods. Our experiments demonstrate that explicit pairing of independently noisy neural SDFs is beneficial in the “no-clean-SDF” paradigm. Thus, our setting is not a convenience for a relatively simple or easy problem, but a principled way to address the challenging no-clean-SDF problem.
>
> We hope this addresses the reviewer’s concern and clarifies both the necessity and the distinctiveness of our problem setup.
>
> # Q2: real-world experiments
>
> We thank the reviewer for the suggestion to further validate our our hypothesis in real-world LiDAR settings. To address this concern, we implemented a LiDAR simulator that mimics a rotating range sensor capturing multiple scans of the same underlying 3D object at different timestamps. Each simulated scan performs perspective projection, applies a z-buffer to retain only the closest return per ray, and injects depth-dependent range noise, random dropout, and outlier points—closely approximating the noise characteristics of real LiDAR measurements. We test three levels of corruption, where higher levels correspond to stronger noise, higher dropout probability, and more outliers, and use the ground-truth mesh to report the Chamfer distance and F1 score. The results demonstrate that our method is possible to generalize to LiDAR-style sensor noise conditions.
>
> | Corrupted Level | **CD (↓)** |        | **F1 (↑)** |        |
> |-----------------|------------|--------|------------|--------|
> |                 | 3DS2V      | Ours   | 3DS2V      | Ours   |
> | **Low**         | 0.016      | 0.012  | 0.961      | 0.990  |
> | **Mid**         | 0.031      | 0.023  | 0.772      | 0.878  |
> | **High**        | 0.059      | 0.052  | 0.518      | 0.601  |

---

> ### Comment · Reviewer_aqUU · 2025-11-26
> **Following questions about the rebuttals.**
>
> Thanks the authors for the concise rebuttals.
>
> Unlike the authors' claims, previous paper [1] is highly similar to the proposed method.
> - First, [1] also utilize the concept of _Noise-to-Noise_.
> - Second, [1] does not rely on clean SDF for supervision.
> - Last, as far as my understanding, [1] also does not require per-scene optimization.
>
> Based on my understanding, may I ask the clear difference between [1] and the proposed submission?
> __This submission and [1] look highly similar.__
>
> Additionally, I found that the authors description about [1] (=Neural Pull) in the related works, which says
>
> > _Gradient regularization techniques like ..., and Neural-Pull (Ma et al., 2020) improve stability and detail._
>
> So, I wonder about the technical difference between [1] and the proposed method. Rest of the issues are quite well resolved throughout the rebuttals. If the addressed problem is resolved, I will change my score.
>
> References
> [1] Learning Signed Distance Functions from Noisy 3D Point Clouds via Noise to Noise Mapping, ICML 2023

---

> > ### Author Response · Authors · 2025-11-27
> > **Key Difference between N2NM and NoiseSDF2NoiseSDF**
> >
> > We thank the reviewer for drawing attention to [1], which we cited as N2NM (Ma et al., 2023) in our paper. We fully agree that N2NM is related in spirit to our work, as both methods draw inspiration from the Noise2Noise paradigm and do not rely on clean SDF supervision. Below we articulate the key technical differences.
> >
> > • Domain of Noise2Noise: point clouds vs neural SDF field.
> > N2NM performs Noise2Noise in the unstructured point cloud domain, where no explicit point-to-point correspondence exists between different noisy observations of the same object. As shown in our Figure 1(b), noisy point clouds are unordered and spatially inconsistent. In contrast, our method performs Noise2Noise directly in the neural field domain, specifically the SDF. As illustrated in Figure 1(c), neural fields preserve coordinate-to-coordinate correspondence across noisy SDFs of the same shape. Therefore, the domain in which Noise2Noise is applied is fundamentally different in N2NM and in our approach.
> >
> > • Training paradigm: overfitting-based vs. data-driven.
> > We would like to clarify that N2NM is widely regarded as an overfitting-based method (for example, PPSurf [a] and the survey [b] explicitly categorize N2NM as an overfitting-based method). N2NM learns the SDF for each individual shape or scene from its own multiple noisy point-cloud observations. In practice, it is trained from scratch for every single shape, which leads to long inference times (around 46 minutes per shape), limited scalability, and no ability to generalize across different shapes. In contrast, our NoiseSDF2NoiseSDF is data-driven: we train a single neural SDF model over a collection of shapes, and once trained, it can be applied to unseen shapes using only a single noisy point cloud in a single forward pass, enabling fast inference (around only 0.05 seconds) without per-shape optimization.
> >
> > • Loss function: EMD vs MSE.
> > Because N2NM operates in the point-cloud domain without explicit point-to-point correspondences, it has to introduce an Earth Mover’s Distance (EMD) loss to establish only soft geometric correspondences between point sets, which is computationally expensive and approximate. In our neural-field setting, coordinates are inherently aligned across SDFs of the same shape, which allows us to use a simple and effective MSE loss between SDF values, without EMD, without point-set association, and without any explicit geometric correspondence search.
> >
> > **Summary of Key Differences**
> >
> > | Aspect                 | **N2NM (Ma et al., 2023)**              | **NoiseSDF2NoiseSDF (Ours)**           |
> > | -------------------------- | --------------------------------------------------- | ------------------------------------------------------------------------ |
> > | **Domain**                 | Operates on **point-cloud data** (unordered sets of 3D points).                                              | Operates on **neural fields**—a continuous function that outputs a Signed Distance Field (SDF).                                                |
> > | **Training Paradigm**        | **Overfitting-based model**: trains a separate model for each single object to fit its SDF.                                 | **Data-driven generalizable model**: one model is trained on many shapes and can predict SDFs for unseen ones without per-object training. |
> > | **Supervision Signal**     | Uses **noisy point clouds** as input and target (point → point).                                                 | Uses **noisy neural SDFs** as input and target (SDF → SDF).                                                                                    |
> > | **Loss Function**          | Expensive **Earth Mover’s Distance (EMD)** to compare unordered point sets. | Simple **Mean Squared Error (MSE)** on SDF values at arbitrary 3D coordinates |
> > | **Inference-Time Optimization** | **Needed**: requires many noisy point clouds and iterative optimization to fit an SDF per object.                         | **Not needed**: a single forward pass directly outputs a clean SDF for an unseen shape from one noisy observation.                             |
> > | **Role of Noise2Noise**    | Applies Noise2Noise **in point space** using **soft** correspondences between noisy point clouds.                         | Applies Noise2Noise **in continuous SDF space**, exploiting the implicit field’s **coordinate-to-coordinate** correspondence                                         |
> >
> > a. Ppsurf: Combining patches and point convolutions for detailed surface reconstruction. CGF 2024.
> >
> > b. Deep Learning For Point Cloud Denoising: A Survey. arXiv:2508.11932.

---

> > > ### Author Response · Authors · 2025-11-27
> > >
> > > "Additionally, I found that the authors description about [1] (=Neural Pull) in the related works"
> > >
> > > We would like to clarify that the paper the reviewer refers to as [1] is, in our manuscript, cited as "N2NM (Ma et al., 2023)":
> > >
> > > [1] Learning Signed Distance Functions from Noisy 3D Point Clouds via Noise to Noise Mapping, ICML 2023.
> > >
> > > In the related work section of our paper, the description the reviewer quoted is not about this 2023 work, but about an earlier and distinct paper from the same group, which we cite as "Neural-Pull (Ma et al., 2020)" [2]:
> > >
> > > [2] Neural-pull: Learning signed distance functions from point clouds by learning to pull space onto surfaces, ICML 2021.
> > >
> > > Both papers share the same first author and have similar citation formats, which may have caused the confusion. In our manuscript, when we discuss the work [1], we refer it as "N2NM (Ma et al., 2023)", not "Neural-Pull (Ma et al., 2021)".
> > >
> > > In addition to elaborating the differences between our work and N2NM [1] as above, we would also like to show that Neural-Pull [2] is a classical \emph{per-scene overfitting} method that learns an SDF from a single input point cloud using its developed gradient-based pulling mechanism. In contrast, our NoiseSDF2NoiseSDF is neither an overfitting method nor applying such a "pulling" mechanism.
> > >
> > > We hope these explanations address the reviewer’s concerns and help clarify possible confusions.

---

### Author Response · Authors · 2025-12-03
**Final Discussion Summary**

Dear Area Chairs,

Our NoiseSDF2NoiseSDF learns clean neural SDFs from noisy supervision, achieving strong performance, fast inference, and robust generalization. To the best of our knowledge, it is the first work to formulate and validate a Noise2Noise training hypothesis directly in neural SDF space. The reviewers have highlighted strengths:

- **Original idea / conceptual novelty** – the core insight is described as novel and impactful (wKfe) and elegant and conceptually novel (ZSTJ).
- **Experimental strength** – solid experimental results and meaningful comparisons (ZSTJ, gKoz).
- **Practicality** – a practical method that does not require clean SDF supervision (ZSTJ).
- **Presentation quality** – the paper is well-written (aqUU, wKfe), with a clear pipeline description and effective visualizations (wKfe).

Reviewer feedback overview:
- Reviewer aqUU (Rating 2, with lowest Confidence 3): explicitly indicated an intention to update the rating (“I will change my score”)
- Reviewer wKfe (Rating 6).
- Reviewer ZSTJ (Rating 6).
- Reviewer gKoz (Rating 6): confirmed "my decision as a weak accept".

We provided comprehensive responses to all reviewers’ comments, and we believe the main concerns have been successfully addressed:
For Reviewer aqUU:
- Regarding the "problem setup", We clarified that our method is a **data-driven** neural SDF model that is trained without access to clean SDF supervision, in contrast to prior **per-shape overfitting** and **Noise2Clean** approaches that require **clean SDF supervision**.
- Regarding the "technical difference between [1]", we explicitly contrasted our work with **N2NM (Ma et al., 2023)** in terms of **domain** (point space vs. neural SDF field), **training paradigm** (overfitting vs. data-driven), and **loss** (EMD vs. MSE),
- Regarding "about [1] (=Neural Pull)", we clarified the distinction between the citations of N2NM and Neural-Pull in the Related Work section.
- Regarding "real-world experiments", we added LiDAR-style simulated experiments with depth-dependent noise, dropout, and outliers, showing strong robustness and generalization.

For Reviewer wKfe:
- Regarding “Point2SDF provider”, we systematically analyzed the role of the **Point2SDF** by using multiple backbones, noise levels, and cross-domain settings, consistently showing that our method improves over each provider and remains robust even when the provider is weaker or domain-mismatched.

For Reviewer ZSTJ:
- Regarding the "comparison", we included the suggested overfitting method **Neural-Singular-Hessian** and the traditional **Poisson Reconstruction**, and our method outperforms both methods on the reported metrics.
- Regarding the "multiple noisy-to-noisy mappings", we evaluated a **multi-target (1-to-3) Noise2Noise** variant, which shows similar final performance and indicates that the original **1-to-1** setting is already effectively sufficient in practice.

For Reviewers wKfe and ZSTJ (shared concerns):
- Regarding the "Real-world sensor noise" and "Robustness analysis", we added experiments with realistic LiDAR-style noise, showing that our method remains robust and accurate beyond Gaussian noise.
- Regarding the "Denoising-Then-Reconstruction comparison", we implemented a denoising-then-reconstruction pipeline, for which our end-to-end neural-field Noise2Noise framework achieves better reconstruction metrics.

For Reviewer gKoz:
- Regarding the "Theoretical justification", we resolved the concern by providing a normal-based local SDF model derivation and a first-order analysis for **approximately unbiased expectations under zero-mean noise**, including at the zero-level set. After examining our analysis and the cited literature, Reviewer gKoz acknowledged that our theoretical justification “can indeed be applied without issue.”
- Regarding the "contribution", We clarified that the main contribution is a Noise2Noise formulation in neural SDF space, a **new training paradigm** for learning clean neural fields from noisy supervision, rather than architectural changes or algorithmic component.
- Regarding the "SDF selected", we justified the use of SDFs over voxel grids via their continuous, coordinate-aligned representation and suitability for high-fidelity surface reconstruction.
- Regarding the "noise level", we showed that the chosen noise levels follow standard practice in prior work.
- Regarding the "beyond prior work", we demonstrated that we are the first to formulate the Noise2Noise directly in neural SDF space.

We thank all reviewers and the Area Chairs for their time, constructive feedback, and thoughtful consideration of our work.

Best regards,
Authors

---

### Meta-Review · Area_Chair_X3YM · 2026-01-06

**Summary:**

The reviewers acknowledge the paper's clear and elegant conceptual extension of the Noise2Noise paradigm to neural SDF fields, its solid experimental validation, and practical utility. However, the final decision leans towards rejection due to a fundamental and unresolved concern regarding the paper's core novelty. The most critical review, from aqUU (score: 2), argues that the problem setup is "weak" and that the method is "highly similar" to prior work (N2NM), essentially constituting an explicit version of an implicit assumption in existing per-scene overfitting methods. While the authors provided detailed distinctions in domain, training paradigm, and loss function, they did not convincingly dispel the perception that the core conceptual leap from 2D images to 3D fields is insufficiently novel or that their formulation represents a significant departure from prior art.

**Reviewer Concerns:**

The rebuttal effectively addressed several technical concerns: it added experiments with LiDAR-style noise and a denoising-then-reconstruction pipeline (for wKfe and ZSTJ), provided additional comparisons (ZSTJ), and engaged in a detailed theoretical justification that satisfied reviewer gKoz's initial doubts. However, the most significant concern from Reviewer aqUU remains outstanding. This reviewer's critique is not about missing experiments or clarifications but challenges the paper's very premise and novelty. The rebuttal clarifies differences with N2NM, but the core of aqUU's argument, that the paper tackles a "simpler" problem by making an implicit practice explicit, and that this does not constitute a substantial novel contribution, was not refuted. This perception that the work is incremental and its conceptual novelty is overstated is a critical, unresolved issue that directly impacts the paper's acceptability.

**Reviewer Scores:**

Reviewer aqUU (Initial: 2) will not change the score since the concerns about novelty were not addressed.

---

### Decision · Program_Chairs · 2026-01-26

Reject